# Ultrafast coupled charge and spin dynamics in strongly correlated NiO

Konrad Gillmeister[1], Denis Golež[2,3], Cheng-Tien Chiang [1], Nikolaj Bittner[3], Yaroslav Pavlyukh [4], Jamal Berakdar [1], Philipp Werner [3✉] & Wolf Widdra [1,5✉]

Charge excitations across an electronic band gap play an important role in opto-electronics and light harvesting. In contrast to conventional semiconductors, studies of above-band-gap photoexcitations in strongly correlated materials are still in their infancy. Here we reveal the ultrafast dynamics controlled by Hund's physics in strongly correlated photoexcited NiO. By combining time-resolved two-photon photoemission experiments with state-of-the-art numerical calculations, an ultrafast ($\lesssim$10 fs) relaxation due to Hund excitations and related photo-induced in-gap states are identified. Remarkably, the weight of these in-gap states displays long-lived coherent THz oscillations up to 2 ps at low temperature. The frequency of these oscillations corresponds to the strength of the antiferromagnetic superexchange interaction in NiO and their lifetime vanishes slightly above the Néel temperature. Numerical simulations of a two-band $t$-$J$ model reveal that the THz oscillations originate from the interplay between local many-body excitations and antiferromagnetic spin correlations.

[1] Institute of Physics, Martin-Luther-Universität Halle-Wittenberg, 06120 Halle, Germany. [2] Center for Computational Quantum Physics, Flatiron Institute, 162 Fifth Avenue, New York NY 10010, USA. [3] Department of Physics, University of Fribourg, 1700 Fribourg, Switzerland. [4] Department of Physics, Technische Universität Kaiserslautern, 67653 Kaiserslautern, Germany. [5] Max Planck Institute of Microstructure Physics, 06120 Halle, Germany. ✉email: philipp.werner@unifr.ch; wolf.widdra@physik.uni-halle.de

Employing light to understand and control ordered states in solid-state systems is a key element of modern science and technology. The optical control of spins in ferromagnets is an especially important research area, related to the ultrafast writing of magnetic information, which has recently been extended to antiferromagnets (AFM)[1]. An interesting class of AFM materials is the family of strongly correlated electron systems, where the local electron correlations lead to insulating behavior and simultaneously stabilize the long-range magnetic ordering via Anderson's superexchange mechanism[2]. The complex electronic structure of correlated materials reflects the coupling mechanisms between the magnetic order and the orbital, spin, and lattice degrees of freedom. These mechanisms lead to numerous competing metastable phases accessible via ultrafast photoexcitation. In recent years, several exciting scenarios for the ultrafast manipulation of ordered states have been reported, including possible high-temperature superconducting states in light-driven cuprates[3,4] and fullerides[5], hidden states in 1T-TaS₂[6], as well as photo-induced non-thermal magnetic and orbital-ordered states[7,8]. Less attention has been paid to the feedback mechanisms between the long-range electronic or magnetic ordering and the dynamics of the photoexcited charge carriers.

In correlated transition-metal oxides (TMO), the strong electron-electron repulsion splits the $d$ bands into an occupied lower Hubbard band (LHB) and an empty upper Hubbard band (UHB). The presence of additional oxygen (ligand $L$) $p$-bands led to the Zaanen, Sawatzky, and Allen (ZSA) classification scheme[9], which suggests that the electronic properties of TMOs are essentially determined by two parameters: the Hubbard $U$ and the charge transfer $\Delta$. Mott-Hubbard insulators are characterized by $U < \Delta$. In these systems, the ligand band is located well below the TM $d$ band and plays a minor role in the low-energy dynamics. In charge-transfer insulators, $\Delta$ is smaller than $U$, and the $p$-band is located between the LHB and the UHB, as schematically depicted in Fig. 1a.

While NiO is a charge-transfer insulator according to this classification scheme, the actual band structure is more complex. This is due to the $d^8$ ground state electronic configuration with formally fully occupied $t_{2g}$ and half-filled $e_g$ orbitals, as indicated in the dark and light blue insets of Fig. 1b, respectively. The established energy scales for NiO are[10,11]:

$$\Delta = E(d^9 \underline{L}) - E(d^8) \sim 4 \text{ eV}, \quad U = E(d^7) + E(d^9) - 2E(d^8) \sim 7.5 \text{ eV},$$

where $\underline{L}$ denotes a hole at the ligand site. Thus, NiO should be considered an intermediate charge-transfer insulator with the $p$-bands located in the energy range of the LHB. The latter situation leads to a strong hybridization between the $2p$ and $3d$ bands and their bonding combination corresponds to the Zhang–Rice doublet. A consistent theoretical description requires a treatment of the correlated $2p$ and $3d$ bands on equal footing as has been demonstrated by Kuneš et al.[12] using the local density approximation (LDA) plus dynamical mean-field theory (DMFT) scheme.

In this work, we explore photoexcited transient states in NiO by exploiting recent advances in photoemission spectroscopy and numerical simulations. More specifically, using pump-probe time-resolved two-photon photoemission (2PPE) with two independently tunable ultraviolet (UV) sources, we study the fate of the excited electrons in the UHB of the transition-metal oxide NiO triggered by photoexcitation across the band gap. Upon above-band-gap optical excitation, we find: (a) an ultrafast decay of the excited electrons within less than 10 fs, and (b) a photo-induced in-gap state well below the bottom of the UHB of the originally undoped system. These two effects represent a non-equilibrium manifestation of the Hund's coupling in NiO. We demonstrate that the many-body in-gap state is coupled to the antiferromagnetic spin background providing conditions for coherent THz oscillations of the 2PPE signal. The

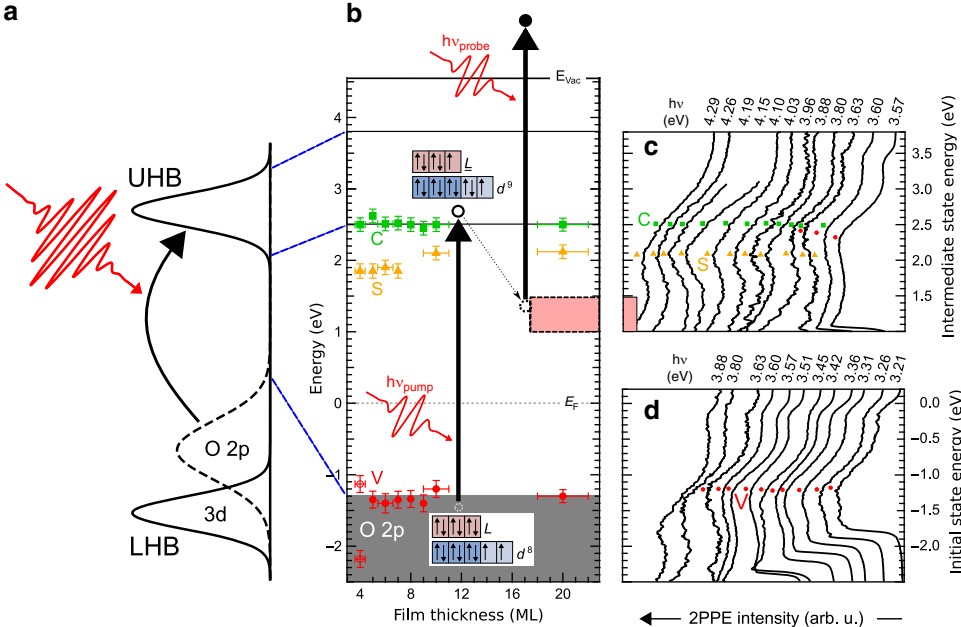

**Fig. 1 Two-photon photoemission of NiO. a** Schematics of photoexcitation in an ideal charge-transfer insulator. LHB (UHB): lower (upper) Hubbard bands formed by the $3d$ electrons; O $2p$: oxygen valence bands. **b** Electronic structure of ultrathin NiO films on Ag(001) with a thickness from 4 to 20 ML. Black arrows illustrate the pump ($h\nu_{pump}$) and probe ($h\nu_{probe}$) processes in time-resolved 2PPE. The relevant electronic states are: $d^8 \underline{L}$ ground states (V, bottom), photoexcited $d^9 \underline{L}$ states of the UHB (S and C, top), as well as photo-induced in-gap states associated with the THz oscillations (red filled). Red boxes of $L$ ($\underline{L}$) in the insets indicate the ligand $2p$ states without (with) a photo-hole, whereas dark (light) blue boxes represent the $t_{2g}$ ($e_g$) orbitals of the $3d$ states. **c, d** 2PPE spectra of 10 ML NiO films for variable photon energies $h\nu_{pump} = h\nu_{probe}$ as indicated above the spectra (laser pulse energies vary between 2 and 50 µJ/cm² depending on the photon energies $h\nu$).

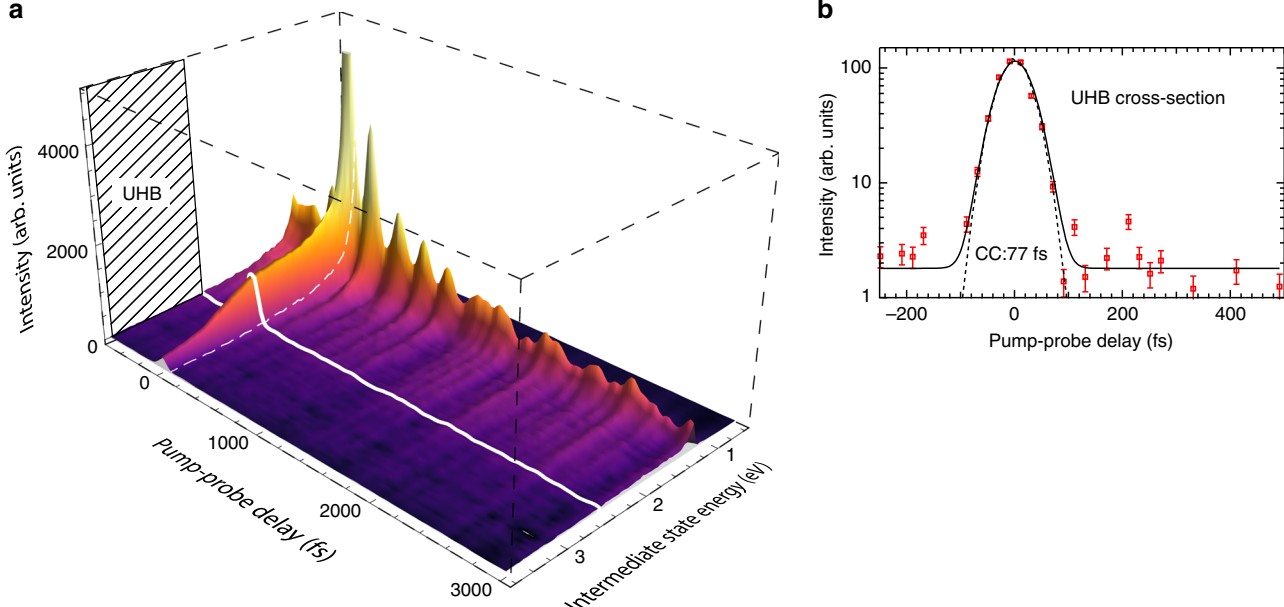

**Fig. 2 Time-resolved pump-probe 2PPE data. a** Experimental data ($h\nu_{pump}$ = 4.2 eV, $h\nu_{probe}$ = 3.4 eV, $T$ = 150 K) for a 9 ML NiO(001) thin film with background subtracted and smoothing using a low-pass filter (without filtering in Fig. 3). The thick white line indicates the position of the lower edge of the UHB at 2.5 eV. **b** 2PPE intensity at the UHB bottom versus pump-probe delay. The cross-correlation (CC) of the pump and probe pulses (dashed line) has been determined from the 2PPE signal at higher energies.

frequency of the THz oscillations corresponds to the super-exchange interaction in NiO, and their amplitude vanishes slightly above the Néel temperature ($T_N$). These experimental findings are complemented by theoretical modeling based on a two-band $t$-$J$ model, which reveals that the coherent oscillations originate from the strong coupling between the excited Hund states and the antiferromagnetically correlated spin background.

## Results

**Time-resolved two-photon photoemission.** The excitation across the optical band gap as studied in the present work is sketched in Fig. 1a, b. The photoexcitation transfers an electron into the UHB, resulting in a hole in the ligand $p$ orbital and a triply occupied site in the Ni $e_g$ manifold (a triplon state). We explore these states and their ultrafast dynamics by a subsequent UV probe pulse, which photoemits the excited electron and projects the whole system to a final state with a ligand hole. Pump-probe 2PPE spectra are shown in Fig. 1c, d for a NiO(001)-(1 × 1) ultrathin film grown on Ag(001) at zero delay between the UV pulses. The 2PPE spectra reveal one occupied state at −1.3 eV with respect to the Fermi level $E_F$, marked as V in Fig. 1b, d, and two unoccupied states at 2.0 and 2.5 eV above $E_F$ marked as S and C, respectively, in Fig. 1b, c. The state V is at the top of the oxygen $2p$ band and is assigned to the Zhang–Rice doublet ($3d^8\underline{Z}$) similarly to the one-hole final state in conventional photoemission experiments[12–14]. The unoccupied states C and S are visible for all film thicknesses and their binding energies depend only weakly on the NiO film thickness for epitaxial films between 4 and 20 ML as summarized in Fig. 1b. The state C is assigned to photoemission from the lower edge of the UHB based on its energy-difference of $\Delta = (3.8 \pm 0.2)$ eV with respect to photoemission from occupied states and the known value of the NiO charge-transfer gap[15]. The state S corresponds to photoemission from an unoccupied $3d_{z^2}$ surface state that has been predicted by theory[16,17]. This state is also visible in surface-sensitive scanning tunneling spectroscopy (STS) as shown in the

Supplementary Note 1. A pronounced STS signal between 1.8 and 1.9 eV above $E_F$ demonstrates its surface state character.

The time-resolved 2PPE spectra are depicted in Fig. 2a for pump-probe delays up to 3 ps. The pump-photon $h\nu_{pump}$ = 4.2 eV promotes the electron into the UHB with a maximum excess energy of about 0.4 eV. The relaxation within the UHB and the population of the in-gap state takes place on ultrafast time scales well below the temporal width of our pump-probe cross correlation, which has been measured to be about 80 fs (Fig. 2b). A detailed analysis of the time traces around zero delay results in an upper limit of 10 fs for the electron lifetime in the UHB (Supplementary Note 1). We emphasize that the observed electron lifetime at the bottom of the (undoped) UHB is three orders-of-magnitude shorter than the value expected at the conduction band minimum of conventional semiconductors with a similar band gap[18–20].

In order to understand the efficient energy dissipation of the UHB electrons, we take a closer look at the 2PPE spectra in the low-energy region. In Fig. 2, we observe a long-lived oscillatory contribution at an intermediate state energy of 1.2 eV, slightly above the low-energy photoemission cut-off. An intensity profile across this energy region reveals an oscillatory component on top of a slowly decaying background (see Fig. 2 as well as Fig. 3a). It is well described by a damped oscillation $I_{osc}(t) = A \cdot e^{-t/\tau} \cdot \cos(2\pi\nu_b \cdot t + \phi)$ with a frequency of $\nu_b = (4.20 \pm 0.26)$ THz, a phase shift $\phi \approx 0$, and a dephasing time of $\tau = (558 \pm 35)$ fs. The oscillation frequency corresponds to $h\nu_b$ =17 meV.

Figure 3a shows the oscillatory signal for the 9 ML ultrathin film in a temperature range from 150 to 455 K. Whereas the oscillation amplitude at $t = 0$ and the oscillation frequency do not vary significantly, the damping of the oscillations increases strongly at higher temperatures. This temperature dependence is summarized in Fig. 3b, d together with the data for a thicker film of 20 ML. In both cases, we find long oscillation lifetimes at low temperatures that can be extrapolated to about 683 and 587 fs at 0 K for 9 and 20 ML, respectively. The temperature dependence of the oscillation lifetime follows the functional form $(1 - T/T')^\nu$, where $\nu$ is the critical exponent for the spin

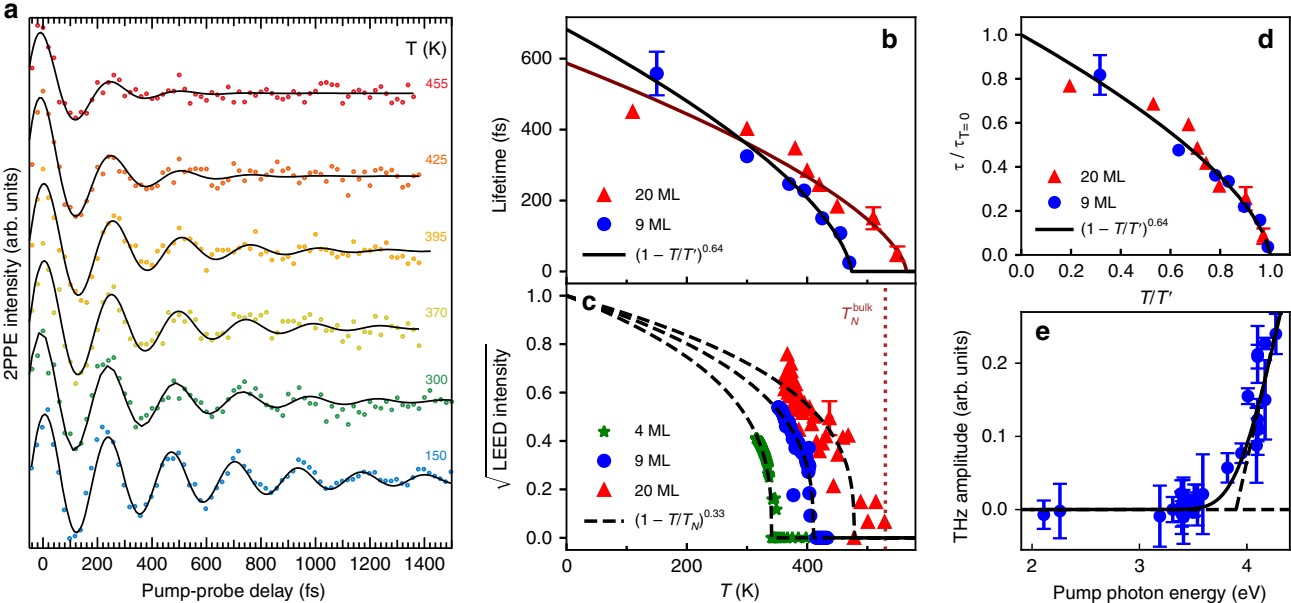

**Fig. 3 Oscillatory contribution of the NiO(001) 2PPE signal for the in-gap state. a** Time-dependent 2PPE signal of a 9 ML NiO(001) thin film for temperatures between 150 and 455 K (1.2 eV intermediate state energy, $h\nu_{pump} = 4.2$ eV with 20 $\mu$J/cm$^2$ and $h\nu_{probe} = 3.4$ eV with 200 $\mu$J/cm$^2$, slowly decaying background subtracted). The black solid lines describe exponentially damped harmonic oscillations. **b** Oscillation lifetime $\tau$ extracted from the time-dependent 2PPE signal for NiO(001) thin films of 9 and 20 ML. The solid line indicates the critical behavior with a critical exponent $\nu = 0.64$, as determined by neutron scattering[21]. **c** Antiferromagnetic scattering amplitude as extracted from the (2 × 1) magnetic superstructure spots in LEED as a function of temperature. Dashed lines indicate the extrapolation to $T_N$ with a critical exponent $\beta = 0.33$ as found in ref. [21]. **d** Relative oscillation lifetimes for 9 and 20 ML on a common reduced temperature scale. **e** Pump-photon energy dependence of the oscillation amplitude at $t = 0$ extracted from the time-dependent 2PPE signal for the NiO(001) thin film of 9 ML at 300 K.

correlation length $\nu = 0.64 \pm 0.03$ extracted from neutron scattering[21]. This observation provides a first hint that the oscillations are connected with spin correlations. The temperature $T'$, where the oscillation lifetime drops to zero, is close to the Néel temperature $T_N$: $T' \simeq 1.15\, T_N$, where $T_N$ has been determined experimentally by the disappearance of the antiferromagnetic (2 × 1) superstructure seen in low-energy electron diffraction (LEED), see Fig. 3c and Supplementary Note 1. We argue that the slightly larger value of $T'$ is due to short-range spin correlations, which persist above $T_N$ and influence the time-resolved photoemission experiments. A similar persistence of short-range magnetic order above the Curie temperature has been observed in ferromagnetic iron[22]. The temperature dependence of the oscillation lifetime shown in Fig. 3b is valid for the 20 ML and the 9 ML film, despite their different $T_N$ and $T'$ (cf. Fig. 3d). Note that the reduction of $T_N$ for the thin NiO films with respect to the NiO bulk value is in agreement with ref. [23]. The observed temperature dependence of the oscillation lifetime with a critical transition close to the Néel temperature points clearly to a dominant role of the local AFM spin ordering and constitutes our main finding.

Whereas the observed amplitude at $t = 0$ is only weakly temperature dependent, it shows a strong threshold behavior with respect to the pump-photon energy $h\nu_{pump}$. The $h\nu_{pump}$ dependence is exemplified in Fig. 3e for the 9 ML film, with a clear onset at $h\nu_{pump} = 3.8$ eV that agrees with the magnitude of the charge-transfer energy $\Delta$. Note that the silver substrate has a plasmon resonance at a similar energy of 3.8 eV, which in principle could lead to plasmonic coupling across the interface[24]. However, we consider an excitation across the interface less likely for two reasons: the substrate plasmon should lead to a peaked response at 3.8 eV[24], whereas the THz oscillations show an onset beginning at 3.8 eV with an increasing excitation cross section with higher energy. Secondly, the oscillations are observed over a

wide range of NiO thicknesses with no indication for an interface-related transport mechanism nor a loss of coherence with increasing film thickness. Therefore, we consider the THz oscillations as triggered by the excitation of an O 2p electron across the charge-transfer gap into the UHB. In the following, we will focus on the electron dynamics activated by the photoexcitation into the UHB, while the lower $t_{2g}$ states will not be considered. On the intermediate state energy scale, the photo-electrons that exhibit THz oscillations originate from the photoemission onset up to 1.5 eV above $E_F$ (Supplementary Note 1), which implies an in-gap state located at around 1 eV below the UHB as marked in Fig. 1b in light red.

**Two-band $t$-$J$ model.** To describe the observed electron dynamics in NiO, we developed a two-band extension of the $t$-$J$ model in the AFM phase, as detailed in "Methods" section. The relevant parameters of this model are the Hund coupling $J_H$ and the exchange coupling $J_{ex}$. In equilibrium, the ground state is dominated by local high-spin states $S = 1$ (S1 in Fig. 5) forming an AFM spin configuration. The remaining local many-body states are illustrated in Fig. 5 and include the triply occupied state (triplon, T), the minority high-spin states (S1$^\flat$) and two types of low-spin $S = 0$ Hund excitations. In the first type, the two electrons occupy two different orbitals with an energy cost of $J_H$ (S0) relative to the ground state, whereas in the second type the same orbital is occupied by two electrons with an energy cost of $3J_H$ (S0$^\flat$). These localized excitations have been the subject of intense experimental[25–27] and theoretical[28–30] investigations. In addition, a spin-flip excitation S1$^\flat$ at energy $zJ_{ex}$ is taken into account, where $z$ denotes the number of exchange-coupled neighbors.

In the calculated equilibrium spectral function (black line in Fig. 4a), subbands of the UHB at energies 4.5 and 6 eV can be identified. These dominant peaks are related to Hund excitations[31]. In addition, we can identify small sidebands which originate

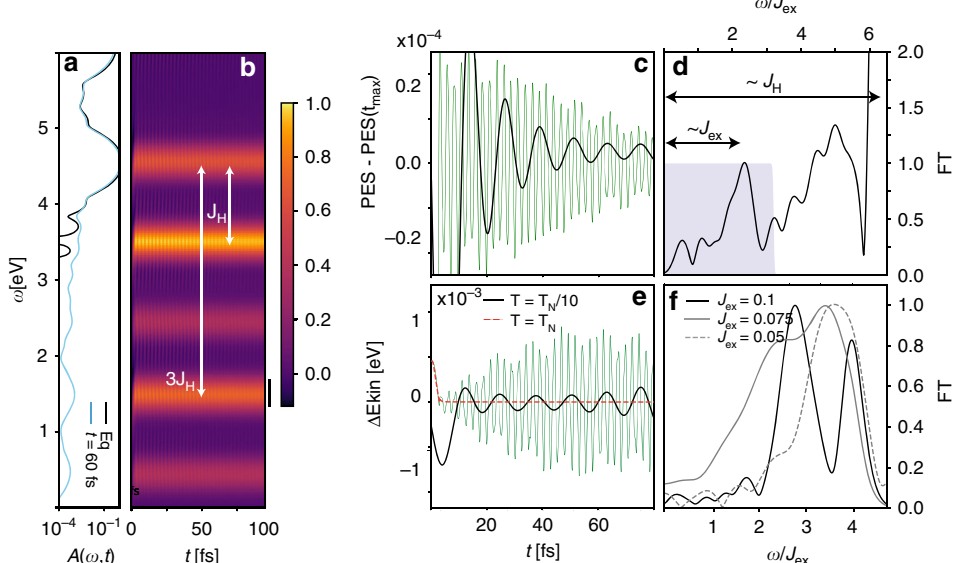

**Fig. 4 Non-equilibrium DMFT calculations for a two-band $t$-$J$ model.** The parameters are $t = 1$ eV, $J_{ex} = 0.1$ eV unless explicitly specified. **a** Spectral function $A(\omega, t)$ in equilibrium (black) and in the photoexcited state (light blue). **b** Theoretical time-dependent photoemission spectrum (PES), see Eq. (7), after the photoexcitation, exhibiting coherent oscillations of the photo-induced in-gap signal. **c** Time evolution of the energy-integrated PES in the window [1.3,1.6] eV, see also the vertical black bar in **b**. The thick black line shows a low-pass filtered signal corresponding to the limited temporal resolution in the experiment, while the thin green lines show the original data exhibiting superimposed rapid oscillations associated with Hund excitations. **d** Normalized Fourier transform of the energy-integrated PES signal exhibiting a low-energy peak scaling with the superexchange interaction $J_{ex}$ and a high-energy peak at an energy proportional to the Hund coupling $J_H$. The gray shaded background corresponds to the employed low-pass filter. **e** Time evolution of the kinetic energy for a temperature below and equal to $T_N$. The thin green line shows the original data and the thick black line the low-pass filtered data. **f** Fourier transform of the kinetic energy for different values of the superexchange interaction $J_{ex} = 0.05, 0.075, 0.1$ eV. In all Fourier transform analyses, the background has been subtracted using a low-order spline interpolation.

from the coupling of charge carriers to the AFM background and the presence of string states[32–36]. To model the pumping by UV excitations across the charge-transfer gap $\Delta$ in the experiments, we excite the system in the theoretical description by photo-doping of electrons near the lower edge of the UHB. During the photo-doping process, triplons T are produced. In the experiments, the photoexcited system is subsequently probed by photoemission, which corresponds to the removal of an electron from the system. The removal of one electron from the triplon state results in a low-spin S0 or S0$^{\flat}$ state, which produces the in-gap features evident in the light blue spectrum in Fig. 4a. The energy splitting of the Hund excitation S0 (S0$^{\flat}$) from the lowest subband of the UHB is given by the Hund coupling $J_H$ (3$J_H$), see white arrows. There is an additional set of in-gap states shifted down in energy by $J_H$, see Supplementary Note 2 for details. As shown in Fig. 4b, which plots the calculated time-dependent photoemission spectrum[37] (see Eq. (7) in "Methods" section), these in-gap states are substantially populated. This analysis allows us to identify the dominant feature in the experiments, shown in Fig. 2a, as the Hund excitation S0$^{\flat}$. In agreement with the experimental findings, all these in-gap features exhibit transient oscillations, which will be analyzed in the following.

The time-resolved photoemission intensity shows dominant fast oscillations associated with Hund excitations. In Fig. 4c, we plot the energy-integrated intensity in the black energy window marked in Fig. 4b by a thin green line. The corresponding Fourier transform, shown in Fig. 4d exhibits a low-energy peak at $\omega$ of the order of $J_{ex}$ and pronounced high-energy peaks near $\omega \approx 0.7 J_H$ and $\omega \approx J_H$. As the fast Hund oscillations cannot be resolved in the experiment, we apply a low-pass filter that extracts the slow oscillations associated with the low-energy peak (see "Methods"

section for details). The slow oscillations are shown by the black line and are found to scale with $J_{ex}$. We thus attribute their origin to the AFM spin correlations in NiO. To further support this interpretation, we compare the time evolution of the kinetic energy below and at $T_N$ in Fig. 4e. Indeed, oscillations with long coherence time only appear at temperatures below $T_N$. As shown in Fig. 4f, the frequency of these oscillations scales approximately linearly with the superexchange $J_{ex}$. We find $\omega/J_{ex} \approx 3$, but cannot exclude lower frequency contributions due to the limited propagation time. The dominant contribution roughly matches the distance between the string states in the two-band $t$-$J$ model. We determined the exchange interaction $J_{ex}$=100 meV by matching the equilibrium $T_N$ with the experiment. This choice leads to exchange interactions which are larger than the realistic values, see "Methods", and thus to faster oscillations in the numerical data.

The initial fast relaxation of the photoexcited triplons happens on a time scale of 10 fs and results from the scattering of triplons with the high-spin states, producing Hund excitations $S = 0$[38]. Since the Hund excitations are mainly determined by the local $J_H$, they are little influenced by the AFM background. As a result, the initial relaxation process in Fig. 4e is similar below and above $T_N$. The initial relaxation is strongly enhanced for high-frequency excitations, where photoexcited electrons are injected up to the upper edge of the UHB. In these processes, the local spin configurations can absorb energy quanta of $J_H$ or $3J_H$ on the time scale of the inverse hopping amplitude $1/t_0$, which is a very efficient dissipation mechanism. This relaxation channel closes once the excess kinetic energy drops below $J_H$, since this is the minimum energy that can be dissipated by local spin excitations. This is the Hund's equivalent of the phonon-window effect[39,40].

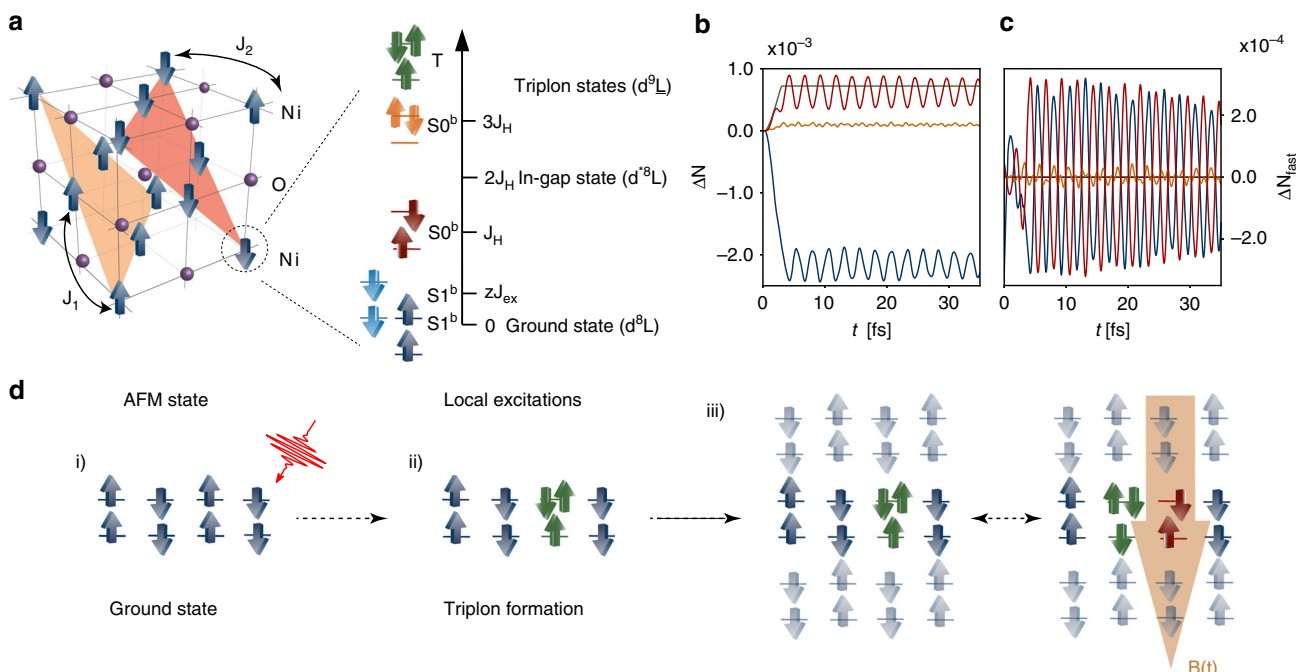

**Fig. 5 Coherent many-body dynamics. a** Schematic view of the NiO lattice structure with FM exchange $J_1$ between the nearest-neighbor spins and AFM exchange $J_2$ between next-nearest neighbors. The local many-body levels are the triplon (T), majority high-spin doublon (S1), minority high-spin doublon (S1$^\flat$), low-spin Hund excitation with electrons in different orbitals at an energy cost $J_H$ (S0), and low-spin Hund excitation with electrons in the same orbital at an energy cost $3J_H$ (S0$^\flat$). Spin-flip excitations such as S1$^\flat$ are described by the exchange Hamiltonian (2). **b** Time evolution of the occupation for the most relevant local many-body states, and **c** the same data with subtracted background dynamics. The colors of the curves match the graphical representation of the local many-body states. **d** Schematic view of the photo-induced dynamics in NiO: (i) AFM ground state with high-spin (S1) states, (ii) photo-induced state with mobile triplons, (iii) coherent dynamics between two many-body states, where the magnetic background $B(t)$ is responsible for the characteristic oscillations whose frequency scales with the strength of the superexchange interaction $J_{ex}$.

## Discussion

Coherent electron dynamics has been observed in isolated quantum systems with few relevant energy levels[41] or for image potential states[42,43]. While these phenomena are related to well isolated quasiparticle physics[44], coherent many-body phenomena are more exotic. Recently, such an observation has been reported in a large array of trapped cold atoms[45]. Various solid-state systems exhibit coherent oscillations, which can emerge for impurity[46] and surfaces states[47,48] or due to the strong coupling to collective[49,50] and phononic modes[51]. However, our work provides the first observation of coherent long-lived many-body dynamics resulting from a coupling to the AFM background in a solid. Similar phenomena have been detected in cuprates exhibiting oscillations with a very short lifetime[52], but their origin remains controversial[53,54].

The experimental observations raise the central question how a strongly correlated solid can support coherent oscillations on such a long-time scale (up to 2 ps). In order to avoid dephasing due to the coupling with a continuum of states, there must exist a small subset of many-body states that are well isolated. The theoretical analysis in Fig. 5b allows us to follow the occupation of the local many-body states in the time evolution. After the photoexcitation, the majority high-spin states (dark blue) are depopulated at the expense of low-spin Hund excitations. The subsequent dynamics is governed by coherent oscillations between the high- and the low-spin states, which can be more clearly seen in Fig. 5c, where the slow background has been subtracted. Since the kinetic energy of the triplons is too small to produce Hund excitations ($E_{kin} < J_H$), the triplons are trapped between nearest-neighbor sites as exemplarily shown in Fig. 5d (iii). This also explains why the in-gap states in the PES are isolated. An analogous process is possible for the high-energy

Hund excitations S0$^\flat$ as well. Furthermore, each creation of a low-spin excitation is accompanied by a ferromagnetic (FM) disturbance in the spin background, which produces a feedback on the spin system in the form of an effective magnetic field $B(t)$ and induces the coherent oscillations at a frequency controlled by the superexchange interaction $J_{ex}$. A string of FM distortions is rapidly created by the pump pulse, which acts as a displacive excitation within the string potential. A natural consequence is a cosine-like response, as evident in the experimental data (Fig. 3), and the simulation results (Fig. 4).

The lifetime of the coherent oscillations is determined by their decay channels. A higher-order process leads to the formation of a FM domain wall and results in a steady increase in the population of the minority high-spin configuration S1$^\flat$ (Supplementary Note 2). This formation of FM strings is one possible relaxation process. However, being a higher-order process with rather low probability, it cannot be responsible alone for the decay of the THz oscillations with a time constant of about $\tau = 500$ fs as observed in the experiments. A spreading of localized FM excitations via magnons can provide an alternative decay channel, which is not captured by the approximate theoretical description considered here and would be an interesting subject for future investigations. The binding of singlons and triplons and its impact on the recombination rate is not properly captured in DMFT, but should not play an important role in the dilute photoexcitation limit explored in the experiments[55]. The experimental observation of oscillations slightly above $T_N$ cannot be explained by our theoretical treatment, which relies on a mean-field decoupling of the spin–spin interaction. It would be interesting to extend the formalism to fluctuating spins[53] in a future study.

In conclusion, we have used two-photon photoemission experiments and a two-band $t$-$J$ model to investigate the

evolution of ultrafast photo-induced in-gap states in strongly correlated NiO. Long-lived coherent oscillations at THz frequencies are observed in the experiments and identified in the theoretical modeling as a signature of the coupling between antiferromagnetically correlated spins and local Hund excitations. Contrary to common belief, this work shows that many-body coherent dynamics can be observed in solid-state systems despite the intrinsic coupling to numerous active degrees of freedom. Our findings indicate an important interplay between local electronic excitations and magnetic order. The combined experimental and theoretical results pave the way to systematic explorations of the many-body physics in correlated oxides driven out of equilibrium by ultrafast above-band-gap excitations. An obvious extension of our work would be the investigation of similar multi-band effects in cold-atom experiments, where two-dimensional antiferromagnetism has been realized recently[56]. The experimental observation of oscillations above the Néel temperature has interesting implications for cold-atom experiments on doped AFMs, where even in the absence of long-range order magnetic correlations have been detected in quantum gas spectroscopy[57].

## Methods

**Theoretical methods.** Crystallographically, NiO is a binary centrosymmetric cubic system of the $m3m$ symmetry class in a paramagnetic state. Each $Ni^{2+}$ ion is surrounded by six $O^{2-}$ ions and has an effective spin $S = 1$. The ligand ions mediate the superexchange interaction between the next-nearest neighbour (NNN) spins (along the $\langle 100 \rangle$ and equivalent directions) resulting in a $J_2 > 0$ exchange constant of the antiferromagnetic type (Fig. 5a). The spin coupling between the nearest neighbour (NN) Ni-ions (along the $\langle 110 \rangle$ and equivalent directions) is much weaker and is of the ferromagnetic type ($J_1 < 0$). Below the Néel temperature, the two exchange interactions result in type-II magnetic order in which the spins of the $Ni^{2+}$ ions are aligned ferromagnetically within {111} planes that form an antiferromagnetic order[26]. The exchange constants $J_1$ and $J_2$ are well-known from experiments and according to Hutchings and Samuelsen[58] are $-1.37$ and 19.0 meV, respectively. The ferromagnetic coupling $J_1$, which is much smaller than the observed oscillation frequency $h\nu_b \approx 17$ meV, is neglected resulting in *four* independent interpenetrating simple cubic lattices[58]. Within each sublattice, there are only the antiferromagnetic interactions between the spins mediated by the ligand $p$-orbitals (predominately a kinetic superexchange which is second order in the hopping between the $Ni^{2+}$ $d$-states and $O^{2-}$ $p$-states[59]). Since we do not treat the superexchange mechanism explicitly, the ligand O-sites are removed from the consideration. Thus, the focus of the theory is a cubic sublattice of spins with AFM nearest neighbour spin-interactions and coordination number $z = 6$.

In the microscopic modeling, we have to consider that each on-site spin $S = 1$ is formed by two elemental spins ($s = 1/2$) located on the respective $e_g$ orbitals. We only retain doubly occupied and triply occupied states (doublons and triplons) and project out the local many-body states which are either unoccupied, singly or fully occupied by introducing the projected operators $\tilde{c}_{i\alpha\sigma} = P^\dagger c_{i\alpha\sigma} P$, where $c_{i\alpha\sigma}$ is the annihilation operator for site $i$, orbital $\alpha$ and spin $\sigma$ and $P$ is the local projector. The resulting Hamiltonian is composed of the local, kinetic and the superexchange parts, $H = H_{loc} + H_{kin} + H_{ex}$. The local part captures the Hubbard and the Hund physics,

$$H_{loc} = U \sum_{i,\alpha} \tilde{n}_{i,\alpha\uparrow} \tilde{n}_{i\alpha\downarrow} - \mu \sum_{i\alpha\sigma} \tilde{n}_{i\alpha\sigma} + \sum_{i,\alpha<\beta} \sum_{\sigma,\sigma'} (U' - J_H \delta_{\sigma\sigma'}) \tilde{n}_{i\alpha\sigma} \tilde{n}_{i\beta\sigma'}$$
$$+ \gamma J_H \sum_{i,\alpha<\beta} \left( \tilde{c}_{i\alpha\uparrow}^\dagger \tilde{c}_{i\alpha\downarrow}^\dagger \tilde{c}_{i\beta\downarrow} \tilde{c}_{i\beta\uparrow} + \tilde{c}_{i\alpha\uparrow}^\dagger \tilde{c}_{i\beta\downarrow}^\dagger \tilde{c}_{i\alpha\downarrow} \tilde{c}_{i\beta\uparrow} \right), \quad (1)$$

where $U$, $U'$, and $J_H$ is the intra-orbital Coulomb, inter-orbital Coulomb, and the Hund's exchange interaction, respectively. The interaction in Eq. (1) is the Kanamori interaction, which becomes rotationally invariant when $U' = U - 2J_H$ and $\gamma = 1$. Here we set $U' = U - 2J_H$ but employ the Ising approximation (with $\gamma = 0$). We have assumed no crystal-field splitting between the two $e_g$ orbitals. The half-filling condition is imposed by adjusting the chemical potential to $\mu = (3U - 5J_H)/2$.

The exchange term reads

$$H_{ex} = J_{ex} \sum_{\alpha \langle ij \rangle} \mathbf{s}_{i\alpha} \cdot \mathbf{s}_{j\alpha}, \quad (2)$$

where $\mathbf{s}$ is a spin-$\frac{1}{2}$ operator, $\langle ij \rangle$ denotes nearest-neighbor indices on the AFM sublattice, and the subscript $\alpha$ labels different $e_g$ orbitals. Due to the presence of long-range order, a mean-field approximation to Eq. (2) is justified, which allows to treat the feedback of the magnetization on the electron dynamics at the Ising level. Comparing the energy cost of a single spin-flip $2zJ_2$ with that given by Eq. (2) in the Ising approximation, $zJ_{ex}$, we infer the relation $2J_2 = J_{ex}$ between the macroscopic and the microscopic exchange parameters.

The formation of AFM spin-order competes with the delocalization of triplon states, with the latter described by

$$H_{kin} = -t_0 \sum_{\langle i,j \rangle} \sum_{\alpha\sigma} (\tilde{c}_{i\alpha\sigma}^\dagger \tilde{c}_{j\alpha\sigma} + \tilde{c}_{j\alpha\sigma}^\dagger \tilde{c}_{i\alpha\sigma}). \quad (3)$$

Here $t_0$ denotes the nearest-neighbor hopping in the AFM sublattice and $\tilde{c}^\dagger$ the (projected) electron creation operator. The competition between the kinetic and the superexchange term results in string-like states at the lower edge of the UHB[32–35].

To solve the electron dynamics, we use the non-equilibrium dynamical mean-field theory[60], which maps a correlated lattice problem onto a self-consistently determined impurity problem. While recent theoretical developments have enabled the description of photo-doped charge-transfer insulators, state-of-the-art simulations are still limited to the paramagnetic phase and short-time dynamics[61,62]. In order to simplify the description, we treat the impurity problem using the projection onto the subspace of relevant local many-body states[63]. In addition, we use the lowest-order strong coupling expansion, the so-called non-crossing approximation (NCA)[53,64]. In order to get access to the long-time behavior, we furthermore solve the problem on a Bethe lattice rather than a cubic lattice, which does not compromise the analysis of the proposed mechanism since the realistic system is not one-dimensional (spin-charge separation does not play a role).

We explicitly break the AFM symmetry and treat the feedback of the AFM order on the mean-field level, see Eq. (2), as

$$J_{ex} \sum_{\alpha \langle ij \rangle} \mathbf{s}_{i\alpha} \mathbf{s}_{j\alpha} \rightarrow J_{ex} \sum_\alpha \sum_i \sum_{jnni} \mathbf{s}_{i\alpha}^z \langle \mathbf{s}_{j\alpha}^z \rangle \quad (4)$$

for both $e_g$ orbitals assuming the spin polarization in the $z$ direction. In the large dimensionality limit, the Hartree term will be dominant and the mean-field decoupling becomes exact. In DMFT, we need to rescale the superexchange as $J_{ex} \rightarrow J^*/z$,[34] where $z$ is the connectivity. The cavity construction, in combination with the mean-field decoupling, leads to

$$H_{ex} = \frac{J^*}{z} \sum_{\alpha \langle i,0 \rangle} \langle \mathbf{s}_{i\alpha}^z \rangle_0 \mathbf{s}_{0\alpha}^z = J^* \sum_\alpha \langle \mathbf{s}_{0\alpha,B}^z \rangle_0 \mathbf{s}_{0\alpha,A}^z = -J^* \sum_\alpha \langle \mathbf{s}_{0\alpha,A}^z \rangle \mathbf{s}_{0\alpha,A}^z, \quad (5)$$

where $\langle \rangle_0$ marks the expectation value in the cavity setup. We have assumed a bipartite lattice with sublattices A and B, and the AFM relation $\langle \mathbf{s}_{0\alpha,A}^z \rangle = -\langle \mathbf{s}_{0\alpha,B}^z \rangle$. In the last step, we have used the assumption of a high-dimensional Bethe lattice to convert the cavity expectation value into a usual lattice expectation value. To roughly match the parameter $J^*$ to the physical $J_{ex}$ in our lattice with $z = 6$, one should choose $(J_{ex}/BW_{NiO}) = (J^*/BW_{Bethe})/6$. With the bandwidths $BW_{NiO} \approx 2–3$ eV and $BW_{Bethe} = 4$ eV, this leads to $J^* \approx 10 J_{ex}$. In practice, we choose $J^* = 0.1$ eV (unless otherwise specified), which also yields a reasonable value for the Néel temperature, $T_{Néel} \approx 500K$. To simplify the discussion of the theoretical results in the main text, we furthermore use the notation $J^* \rightarrow J_{ex}$.

The model parameters other than $J_{ex}$ are taken from equilibrium LDA+U[65] and LDA+DMFT[12,66] studies, which used $U \approx 8$ eV, $J_H \approx 1.0$ eV. We infer the effective hopping $t_0 \approx 1$ eV from the bandwidth of the UHB in the calculation of Kuneš et al.[12] and from the bandwidth of the $d^9$ state in the Bremsstrahlung isochromate spectroscopy[15]. The hopping $t_0$ is of indirect nature and is mediated by the hopping through the oxygen $2p$ states as explained by Zhang and Rice in the case of cuprates[67].

The slow oscillations in Fig. 4 have been extracted by a low-pass filter with a Fermi-function shape $f(\omega) = 1.0/(\exp(\beta(\omega - \omega_0)) + 1)$, see shaded region in Fig. 4d. The presented results are for $J_{ex} = 0.2$ and the parameters $\beta = 100$ and $\omega_0 = 0.8$. As the main limitation in the comparison between the theoretical and experimental results is the maximum propagation time and the corresponding mismatch in the superexchange, we propose this problem as a perfect testbed for future advances in the theoretical modeling of strongly correlated systems out of equilibrium.

In order to model the excitation from a deep fully occupied $p$-band to the UHB, we approximate the excitation by a sudden electron doping in a certain energy range of the UHB. This is justified, since the time scale for the UV excitation at $h\nu = 4.1$ eV corresponds to 0.1 fs according to Heisenberg's uncertainty principle. The UV pump pulse may thus be modeled within the rotating wave approximation exciting electrons from fully occupied orbitals. In all simulations, we have attached the full bath at the lower edge of the UHB in the energy range $\omega = [3, 4]$ eV for the time interval $[0, 2]$ fs. The time-dependent spectral function $A(\omega, t)$ is obtained from the retarded component of the local Green's function $G^R$ as

$$A(\omega, t) = -\frac{1}{\pi} \text{Im} \int_t^{t+t_{cut}} dt' e^{i\omega(t'-t)} G^R(t', t), \quad (6)$$

and the photoemission spectrum is obtained from the lesser component $G^<$ as

$$PES(\omega, t) = \frac{1}{\pi} \text{Im} \int_t^{t+t_{cut}} dt' e^{i\omega(t'-t)} G^<(t', t). \quad (7)$$

Due to the limited propagation time, we have employed a forward integration for the Fourier transform in contrast to ref. [37], where the Fourier integral has been done in relative time.

**Experimental methods**. The experiments have been performed in an ultrahigh vacuum chamber equipped with a 150 mm hemispherical electron analyzer (Phoibos 150, SPECS, Berlin) with a 2D CCD detector and low-energy electron diffraction optics as described in detail elsewhere[68,69]. A broadly tunable femtosecond laser system operated at a repetition rate of 1.4 MHz with two noncollinear optical parametric amplifiers (NOPA), pumped by a 20 W fiber laser (IMPULSE, Clark-MXR, Dexter) is used for the 2PPE[68]. The two frequency-doubled beams of the NOPAs serve as pump-probe excitation source with typical pump-probe cross-correlation widths between 70 and 100 fs (full-width-at-half-maximum).

NiO thin films have been prepared by reactive Ni deposition at room temperature on a Ag(001) single crystal in $10^{-6}$ mbar $O_2$. Subsequent annealing is applied to achieve long-range order as monitored by LEED and high-resolution electron energy loss spectroscopy[70].

## Data availability

The data that support the plots and other findings presented in this paper are available from the corresponding author upon reasonable request.

## Code availability

The numerical code used to calculate the results for this work is available from the corresponding author upon reasonable request.

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

## Acknowledgements

The calculations have been performed on the Beo04 cluster at the University of Fribourg and on the Rusty cluster at the Flatiron Institute using a software library developed by M. Eckstein and H.U.R. Strand. The Flatiron Institute is a division of the Simons Foundation. K.G., C.-T.C., and W.W. acknowledge financial support from the German Research Foundation (Deutsche Forschungsgemeinschaft, DFG) through SFB 762 (A3, B8) and SFB/TRR 227 (A06). D.G., N.B. and P.W. were supported by Swiss National Science Foundation grant 200021-165539 and the European Research Council through ERC consolidator grant No. 724103. Y. P. acknowledges support of the DFG via SFB/TRR 173. D.G. would like to acknowledge R. Sesny for help with graphical representations.

## Author contributions

K.G. and W.W. designed the experiments. K.G. prepared the samples and measured the 2PPE data. The data analysis and evaluation was done by K.G., C.-T.C., Y.P., and W.W., D.G., N.B., Y.P. and P.W. constructed the model description and D.G. carried out the DMFT calculations. All authors discussed the results and co-wrote the paper.

## Competing interests

The authors declare no competing interests.
