## [Peer Review File · Nature Communications]

Reviewers' Comments:

Reviewer #1:

Remarks to the Author:

Gillmeister et al. present ultrafast photoemission measurements of NiO finding a novel oscillatory signal associated with the antiferromagnetic order. This represents a novel type of ultrafast dynamics in a correlated material. Overall, I regard the work to be sufficiently important and well-supported by the data and theory to justify inclusion in Nat. Comm. On the less positive side, there are many points in the manuscript, which I think require improvement in order to make the work publishable.

1. The terms describing magnetic exchange seem to be defined in ways that are inconsistent. Is J_{ex} the energy cost of breaking the strongest magnetic bond or the energy cost of flipping a spin? The text seems to imply the usual definition i.e. J_{ex} is the cost of breaking a single bond, but the diagram in the right panel of Fig 5a implies that J_{ex} is the cost of creating the state with a flipped spin. Fig 5 also implicitly defines J_1 in a way that seems inconsistent with the text. I thought J_1 was the rather weak nearest-neighbor exchange? Not the stronger 2nd neighbor exchange.
2. Somewhat related there is an inconsistency between the theory and experiment, which see oscillations at $\sim J_{ex}$ and $3 \times J_{ex}$ respectively. The suggestion that theory and experiment will merge for smaller values of J_{ex} is not immediately plausible to me given how the theory scales with J_{ex} in the higher limit of J_{ex} in Fig. 4f. I think this issue would be better treated by either: (A) Explaining the reasoning behind this suggestion or (B) Presenting this issue as an unsolved problem that needs further work.
3. I find that the very important comparison between the temperature-dependent decay timescale and T_N is not treated adequately. The plot appears to be designed to test the hypothesis that the time-scale changes like $\sim \sqrt{1-T/T_N}$. This would only really make sense if one assumes that the AFM order parameter also scales like $\sim \sqrt{1-T/T_N}$, but there is no information that this is the case. I think the meaningful comparison is to directly plot the timescales and \sqrt{I} against temperature where I is the intensity of the magnetic scattering the authors currently only present in the supplementary. It is crucial to consider errorbars in both temperature and time in order to convey the statistical significance of the results.
4. The reasoning for the assignment of features V, C and S in Fig 1 should be explained. If it is based on a comparison to the literature, it would be better to cite specific experimental references rather than a wide range of different papers.
5. The way in which Mott-related physics is introduced in the manuscript blurs the distinction between a charge-transfer and Mott-Hubbard type insulator in a way that is rather unhelpful for the readability of the manuscript. It first discusses 2PPE in NiO as involving LHB to UHB transitions before clarifying the fact that the relevant transition is O 2p to UHB. It would be better to be more careful and precise from the outset about when one is talking about a generic Mott insulator and when one is talking about NiO. The second sentence of the Fig. 1 caption is also ambiguous. Does the LHB refer to a composite O 2p + Ni d state or to just the lower energy Ni 3d states? I would not object to either notation, but "LHB" should have one clear meaning in the manuscript.
6. What I think is the same state seems to be referred to using multiple inequivalent descriptions/notations e.g. "hole in the ligand p state = holon = $3d^8L^{-1}$ = $3d^8Z^{-1}$ ". This makes it harder to follow exact meanings, especially as "holon" is more commonly used to refer to a spineless quasiparticle in 1D physics
7. The literature cited all seem to use L to denote a ligand hole. It's not clear why L^{-1} is used here. Right now, the ligand and Ni states have different notation where the superscript is the change in electrons and total electrons, respectively.
8. The meaning of the term "One color" in Figure 1 might not be immediately clear for all readers.
9. The use of colored letters in Fig 1 is inconsistent. The color of the text label sometimes matches the color of the points, but not always.
10. Information about the pump fluence is should be in the main manuscript rather than just in the supplementary.

Reviewer #2:

Remarks to the Author:

Gillmeister et al perform ultrafast 2PPE experiment on NiO. They observe coherent THz oscillations, which they interpret in terms of Hund's physics of spin interactions associated with electron injection into the eg symmetry orbitals. The experiment and theoretical analysis are well done, so I recommend publication. Such physics have not been observed in the time domain, as far as I am aware, and I believe it will be of broad interest to those investigating correlated materials in the time domain. It is quite sophisticated that the experiments are accompanied by high level theory. The authors may consider a few comments.

- 1) T_{2g}-eg transitions appear to be common in transition metal oxides, as suggested by observation of similar excitations in TiO₂.¹ Presumably, the electron lifetimes and the energy level structure has much influence on whether such oscillations can be expected.
- 2) The authors see a temperature dependence of the oscillations that they interpret as the Neel temperature. They should also consider how electron-phonon interaction affects the eg orbitals in transition metals.
- 3) The experimental observables occur at 3.8 eV and are interpreted in terms of the electronic structure of NiO. This is probably correct, and can be supported by the NiO film thickness dependence. The authors, however, do not emphasize how their results depend on the NiO film thickness. This is important, because 3.8 eV also corresponds to the bulk plasmon in Ag(100) substrate,² so they could have a process, where a bulk plasmon is excited and transfers an electron to NiO.³ This certainly happens for Ag/TiO₂ interfaces. Moreover Ag has interband transitions that start at this energy.⁴ For example, Ag d-sp band transitions occur at and above this energy, and could have an influence on the NiO dynamics.

1 Argondizzo, A. et al. Ultrafast Multiphoton Pump-Probe Photoemission Excitation Pathways in Rutile TiO₂. Phys. Rev. B 91, 155429 (2015).

2 Reutzler, M., Li, A., Gumhalter, B. & Petek, H. Nonlinear Plasmonic Photoelectron Response of Ag(111). Phys. Rev. Lett. 123, 017404, (2019).

3 Tan, S. et al. Plasmonic coupling at a metal/semiconductor interface. Nature Photon 11, 806-812, (2017).

4 Rosei, R. Temperature modulation of the optical transitions involving the Fermi surface in Ag: Theory. Phys. Rev. B 10, 474 (1974).

Reviewer #3:

Remarks to the Author:

The manuscript by Gillmeister et al. describes a combined experiment and theoretical study of the photodoping and subsequent relaxation of NiO thin films. The data show long-lived excitations which live inside the gap. The description of this behavior is that of strongly correlated dynamics. The most compelling results in the data are how the decay time of the oscillations scales with the temperature.

Overall, I like the paper and lean toward recommending it for publication. However, I found much of the theory discussion and the integration of the theory with the experiment to be quite confusing. I hope this can be clarified in the next round.

To start, the first part of the manuscript discusses a clear charge-transfer insulator with clear upper and lower Hubbard bands and the charge-transfer band. The ensuing discussion explains how the p-holes become correlated (Zhang-Rice physics) but later we are told they are integrated out because they are noninteracting.

I found it very difficult to reconcile Figure 4 with the experimental results. How exactly am I to do

that? The equilibrium spectra seemed to have far more bands and gaps than the earlier discussion described. The nonequilibrium result had additional structure, but I was confused by this because I thought it was shown after the pump is off. By that point, the retarded Green's function should have returned to its equilibrium form because it rapidly does this (due to the fact that it represents the available quantum states and there is no dressing when the field is off). But this one has additional structure (I am guessing due to the occupancy in the UHB, but the explanation for this was not so clear).

The oscillations shown in Fig. 4 were very rapid. Indeed, they would never be seen in a real experiment because the probes have short time width envelopes, but this physics is not included in the calculations if I followed them properly.

So, for me, there seems to be a huge disconnect between the actual experiments and the theory that is supposed to illustrate the phenomena seen in the experiment. I hope this can be properly remedied, as I did like the paper on the whole.

My recommendation is to reconsider after the manuscript is revised.

Reviewer #1 – NCOMMS-19-1124742-T

Gillmeister et al. present ultrafast photoemission measurements of NiO finding a novel oscillatory signal associated with the antiferromagnetic order. This represents a novel type of ultrafast dynamics in a correlated material. Overall, I regard the work to be sufficiently important and well-supported by the data and theory to justify inclusion in Nat. Comm. On the less positive side, there are many points in the manuscript, which I think require improvement in order to make the work publishable.

We thank the Referee for his/her detailed suggestions and the positive assessment of our work. In the revised version, our manuscript was modified as to address these remarks. Please, find below our detailed responses to the specific questions.

1. The terms describing magnetic exchange seem to be defined in ways that are inconsistent. Is J_{ex} the energy cost of breaking the strongest magnetic bond or the energy cost of flipping a spin? The text seems to imply the usual definition i.e. J_{ex} is the cost of breaking a single bond, but the diagram in the right panel of Fig 5a implies that J_{ex} is the cost of creating the state with a flipped spin spin. Fig 5 also implicitly defines J_1 in a way that seems inconsistent with the text. I thought J_1 was the rather weak nearest-neighbor exchange? Not the stronger 2nd neighbor exchange.

We thank the referee for the very careful reading of our manuscript. Indeed some additional explanations and consistency checks were needed. The picture is as follows: crystallographically, NiO is a binary centrosymmetric cubic system of the $m3m$ symmetry class in a paramagnetic state. Each Ni^{2+} ion is surrounded by six O^{2-} ions and has an effective spin $S = 1$. The ligand ions mediate the superexchange interaction between the next nearest neighbour (NNN) spins (along the $\langle 100 \rangle$ and equivalent directions) resulting in a $J_2 > 0$ exchange constant of the antiferromagnetic type. The spin coupling between the nearest neighbour (NN) Ni-ions (along the $\langle 110 \rangle$ and equivalent directions) is much weaker and is of the ferromagnetic type ($J_1 < 0$). Below the Néel temperature (which for the bulk samples is given by $T_N = 523 \text{ K}$), the two exchange interactions result in a magnetic order in which the spins of the Ni^{2+} ions are aligned ferromagnetically along the $\langle 11\bar{2} \rangle$ axis within $\{111\}$ planes. These planes form an antiferromagnetic order [1]. The exchange constants J_1 and J_2 are well-known from experiments, -1.37 and 19.0 meV , respectively [2, 3]. They are now explicitly indicated in Fig. 5(a), and the above explanations have been added before introducing Eq. (1).

Next we need to connect this realistic picture to the theoretical model. In a first step, the ferromagnetic coupling J_1 , which is much smaller than the observed oscillation frequency $\hbar\nu_b \approx 17 \text{ meV}$, is neglected, which effectively decouples the lattice into *four* interpenetrating unconnected simple cubic sublattices. Within each sublattice, only the antiferromagnetic interactions between the spins remain. Since we do not treat the superexchange mechanism explicitly, the ligand O-sites are removed from the consideration. Thus, our model describes a cubic lattice of spins with Ni–O–Ni exchange pathways becoming the nearest neighbour ($z = 6$) spin-spin interactions. In a second step, we couple the spin order to the electron dynamics by realising that each on-site spin is formed by two elemental spins ($s = 1/2$) located on the respective e_g orbitals. This is now clarified in the completely restructured “Theoretical Methods” part, which contains the model derivation.

The parameter J_{ex} in the model describes the interaction between elemental spins ($s = 1/2$), see Eq. (1) in the main text. For reasons of computational efficiency, we employ a

Bethe lattice in the simulations. By matching the equilibrium Néel temperature to the experimental value, we have determined the exchange interaction $J_{\text{ex}}=100$ meV. This is larger than the realistic exchange interaction J_{ex} , but in any case the limited propagation times do not allow us to analyze photoinduced oscillations for substantially lower exchange couplings.

The microscopic parameter J_{ex} can be connected to the experimentally relevant J_2 by comparing the energy associated with the breaking of a single spin bond and using the Ising approximation $\sum_{\langle ij \rangle} \hat{\mathbf{s}}_i \cdot \hat{\mathbf{s}}_j \rightarrow \sum_{\langle ij \rangle} \hat{s}_i \cdot \hat{s}_j$, which is well justified in the scenario we consider [4]:

$$\begin{aligned} E_b &= J_2 \times 1 \times 1 = J_{\text{ex}} \times 2 \times \frac{1}{2} \times \frac{1}{2}, \\ E_b &= J_2 = \frac{1}{2} J_{\text{ex}}. \end{aligned}$$

Hence, the energy associated with a single spin-flip is given by

$$E_f = 2zJ_2 = zJ_{\text{ex}}.$$

This is now explicitly indicated in Fig. 5(a) and the relation between J_2 and J_{ex} is discussed immediately after Eq. (2).

2. Somewhat related there is an inconsistency between the theory and experiment, which see oscillations at J_{ex} and $3xJ_{\text{ex}}$ respectively. The suggestion that theory and experiment will merge for smaller values of J_{ex} is not immediately plausible to me given how the theory scales with J_{ex} in the higher limit of J_{ex} in Fig. 4f. I think this issue would be better treated by either: (A) Explaining the reasoning behind this suggestion or (B) Presenting this issue as an unsolved problem that needs further work.

We thank the referee for this important question. From a simple theory of doped antiferromagnets in the Ising limit, we expect that this scaling is not linear, but rather a power law, namely $J_{\text{ex}}^{2/3}$, see Refs. 24 and 26. However, from the available data (limited by the propagation time), it is very hard to make a conclusive statement. Therefore, we decided to present the issue as an unsolved problem, which is now mentioned in the methods part as an important theoretical question.

While in the future the computer memory will grow, we expect that better algorithms to solve the nonequilibrium DMFT equations for longer times will be developed. Indeed, there are indications that the generalized Kadanoff-Baym Ansatz for symmetry broken states [5, 6] or memory truncation techniques [7] may be helpful in reducing the complexity of two-time Kadanoff-Baym equations. This work provides a clear example that these efforts are essential for future comparisons with experimental setups. We have added a comment in the main text (theoretical methods):

“As the main limitation in the comparison between the theoretical and experimental results is the maximum propagation time and the corresponding mismatch in the superexchange we propose this problem as a perfect testbed for future advances in theoretical modeling of strongly correlated systems out of equilibrium.”

3. I find that the very important comparison between the temperature-dependent decay timescale and T_N is not treated adequately. The plot appears to be designed to test the hypothesis that the time-scale changes like $\sqrt{1-T/T_N}$. This would only really make sense if one assumes that the AFM order parameter also scales like $\sqrt{1-T/T_N}$, but there is no information that this is the case. I think the meaningful comparison is

to directly plot the timescales and \sqrt{I} against temperature where I is the intensity of the magnetic scattering the authors currently only present in the supplementary. It is crucial to consider error bars in both temperature and time in order to convey the statistical significance of the results.

We thank the referee for suggesting a refined analysis of the temperature dependence with respect to the magnetic scattering amplitude. Remarkably, this analysis has provided two new and important insights:

1. **Critical exponent:** A careful analysis of the critical exponent extracted from the relaxation times reveals the scaling $(1 - T/T')^x$, where T' is the critical temperature for the vanishing of oscillation, with $x \approx 0.64 \pm 0.07$. This is in very good agreement with the critical exponent for the correlation length $\nu = 0.64 \pm 0.03$ extracted from neutron scattering experiments on NiO [8]. Furthermore, the analysis of the LEED intensity of the magnetic scattering shows a critical behavior $(1 - T/T_N)^\beta$, where the critical exponent is $\beta = 0.33$, which also agrees with neutron scattering. This further supports our theoretical analysis that spin correlations are playing an important role in the coherent oscillations. We have replaced the previous exponent corresponding to the mean-field approximation to the XY model in 3 dimensions.
2. **Nonthermal critical point:** A careful comparison of the Néel temperature T_N extracted from the intensity of the magnetic scattering and the critical temperature for the decaying oscillations T' shows a systematic difference: $T' \approx 1.15T_N$. This is an important observation as coherent oscillations are a measure of local spin correlations, while the intensity of the magnetic scattering measures the long-range order. This means that there exists a regime, where the long-range order is destroyed, yet short-range spin correlations are present and can be identified by coherent oscillations in photoemission spectroscopy. In a similar range, magnetic exchange splitting has been observed above the Curie temperature in iron due to local magnetic order [9]. There exists an exciting parallel with chemically doped antiferromagnets, as is now mentioned in the conclusions. Unfortunately, it is very hard to model the regime with short-range correlations with our theoretical tools, and we thus mention this problem as an important future challenge.

The manuscript has been changed accordingly on page 6 and in Fig. 3. The error bars for the lifetime and the LEED intensities are now indicated in the figures. The relative temperature error is about 5 K and the absolute error is estimated to be 25 K. This is now also stated in the Supplementary Information. The citation of the work on the magnetic exchange splitting of iron is included in the manuscript. We also mention a theoretical challenge at the end of the manuscript: “The experimental observation of oscillations slightly above T_N cannot be explained by our theoretical treatment, which relies on a mean-field decoupling of the spin-spin interaction. It would be interesting to extend the formalism to fluctuating spins [10] in a future study.”

4. The reasoning for the assignment of features V, C and S in Fig 1 should be explained. If it is based on a comparison to the literature, it would be better to cite specific experimental references rather than a wide range of different papers.

For the state V, a clear reasoning and a comparison with the available literature data (Refs. 13-15) have been presented in the manuscript already. For the states C and S we extended the discussion as follows:

“The unoccupied states C and S are visible for all film thicknesses and their binding energies depend only weakly on the NiO film thickness for epitaxial films between 4 and 20 ML as summarized in Fig. 1(b). The state C is assigned to photoemission from the lower edge of the UHB based on its energy difference of $\Delta = 3.8 \pm 0.2$ eV with respect to photoemission from occupied states in comparison with the known value of the NiO charge-transfer gap. The state S corresponds to photoemission from an unoccupied $3d_{z^2}$ surface state that has been predicted by theory [Ködderitzsch et al., Phys. Rev. B **66**, 064434 (2002), Schroen *et al.*, J. Phys. Condens. Matter **25**, 094006 (2013)]. This state is also visible in surface-sensitive scanning tunneling spectroscopy (STS) as shown in the Supplementary Information. A pronounced STS signal between 1.8 and 1.9 eV above E_F demonstrates its surface state character.” The STM/STS data have been added to the Supplementary Information.

5. The way in which Mott-related physics is introduced in the manuscript blurs the distinction between a charge-transfer and Mott-Hubbard type insulator in a way that is rather unhelpful for the readability of the manuscript. It first discusses 2PPE in NiO as involving LHB to UHB transitions before clarifying the fact that the relevant transition is O 2p to UHB. It would be better to be more careful and precise from the outset about when one is talking about a generic Mott insulator and when one is talking about NiO. The second sentence of the Fig. 1 caption is also ambiguous. Does the LHB refer to a composite O 2p + Ni d state or to just the lower energy Ni 3d states? I would not object to either notation, but “LHB” should have one clear meaning in the manuscript.

We thank the referee for this feedback. Since in our theoretical description we focus only on the UHB, the distinction between Mott and charge-transfer insulator is irrelevant. However, for a better readability, we have reorganized and partially reformulated our introduction. In particular, we did a careful classification of the transition-metal oxides according to Zaanen, Sawatzky, and Allen. The band structure of NiO and its classification with respect to this scheme is introduced afterwards. The discussion of 2PPE in NiO involving O2p to UHB has been moved to page 4.

The “LHB” corresponds to the lower energy of Ni 3d states. We have checked this notation throughout our manuscript.

6. What I think is the same state seems to be referred to using multiple inequivalent descriptions/ notations e.g. “hole in the ligand p state” = holon = $3d^8L-1 = 3d^8Z-1$. This makes it harder to follow exact meanings, especially as “holon” is more commonly used to refer to a spineless quasiparticle in 1D physics.

We thank the referee for this remark. The hole in the ligand p -state gives rise to 2 broad classes of electronic states:

1. $3d^8\bar{L}$ is a weakly interacting feature in the spectrum. The strongly localized hole represents the dominant final state in photoemission from charge-transfer insulators [Zaanen, Sawatzky, and Allen, Phys. Rev. Lett. **55**, 418 (1985)]. No spin order is required for its observation.
2. $3d^8\bar{Z}$ is a bound screened state. The screening is indicated by the letter Z . This notation originates from atomic physics—the hydrogen-like atoms. However, the screening mechanism (Zhang-Rice) is quite different, being realized through the hopping of spin (rather than charge) in the AFM background. This idea first appeared

in the context of the t - J model. The background spin-order is essential. This class of electronic states is broad because of the fine structure associated with different string states.

In NiO both features appear simultaneously. In fact, early theoretical work of Bała, Oleś and Zaanen, Phys. Rev. B **61**, 13573 (2000) has been confirmed experimentally by Taguchi *et al.*, Phys. Rev. Lett. **100**, 206401 (2008), see figure 1 below.

Indeed the referee is right that the expression “holon” is used for spinless particles in 1D setups. In the theoretical community of ultrafast processes holon is often used to describe an unoccupied state in the Hubbard model after a photo-excitation. To avoid any confusions, we removed the term holon from the manuscript.

Figure 1: Final electronic states in the core and valence band photoemission spectroscopy of NiO, adapted from Taguchi *et al.*, Phys. Rev. Lett. **100**, 206401 (2008).

7. The literature cited all seem to use L to denote a ligand hole. It’s not clear why L-1 is used here. Right now, the ligand and Ni states have different notation where the superscript is the change in electrons and total electrons, respectively.

The original idea was to improve the legibility by replacing the underscore with negative superscript to denote the ligand hole. However, this is indeed unconventional as the referee rightfully pointed out. Therefore, in the new version we implemented the following changes:

$3d^8L$ is used to denote the fully occupied ligand $2p$ -shell.

$3d^8L^{-1}$ was used to denote a hole on the ligand $2p$ -shell. Now, the $3d^8\bar{L}$ notation is used.

8. The meaning of the term “One color” in Figure 1 might not be immediately clear for all readers.

We modified the figure caption of Fig.1 and avoid the term “one-color”.

9. The use of colored letters in Fig 1 is inconsistent. The color of the text label sometimes matches the color of the points, but not always.

We thank the referee for catching this glitch, which we have corrected.

10. Information about the pump fluence should be in the main manuscript rather than just in the supplementary.

The pump and probe laser fluences are specified in the main manuscript (in the figure captions of Fig. 1 and 3).

Reviewer #2 – NCOMMS-19-1124742-T

Gillmeister et al perform ultrafast 2PPE experiment on NiO. They observe coherent THz oscillations, which they interpret in terms of Hund's physics of spin interactions associated with electron injection into the eg symmetry orbitals. The experiment and theoretical analysis are well done, so I recommend publication. Such physics have not been observed in the time domain, as far as I am aware, and I believe it will be of broad interest to those investigating correlated materials in the time domain. It is quite sophisticated that the experiments are accompanied by high level theory. The authors may consider a few comments.

We thank the Referee for his/her careful review and for recommending our work for publication.

1. T_{2g}-eg transitions appear to be common in transition metal oxides, as suggested by observation of similar excitations in TiO₂. Presumably, the electron lifetimes and the energy level structure has much influence on whether such oscillations can be expected.

The situation for the transition metal oxide TiO₂ is quite different since it has no occupied *d* states in the ground state. Therefore, there is no magnetic coupling as in NiO and, e.g., MnO or CoO. At surfaces, there are in-gap states reported for TiO₂, which correspond to a Ti 3*d*¹ configuration. However these are diluted defect states, which are very different from the case of Ni *d*⁸ in NiO. It is therefore not discussed in the manuscript. In addition, one expects that electron correlation effects upon photoexcitation do not play a major role since the Coulomb repulsion between localized electrons in *d*¹ states is absent.

2. The authors see a temperature dependence of the oscillations that they interpret as the Néel temperature. They should also consider how electron-phonon interaction affects the eg orbitals in transition metals.

Vibrational excitations in NiO span a wide energy range from zero to about 70 meV as has been studied for NiO thin films earlier by us (see Refs. Kostov *et al.*, Phys. Rev. B **87**, 235416 (2013) and Phys. Rev. B **94**, 075438 (2016)). Although the energy of 17 meV of the observed THz oscillation falls in this energy range, there is neither a specific phonon mode nor a higher density-of-states feature close to this energy. In addition, the specific temperature dependence of the THz oscillation points clearly to an antiferromagnetic origin. One might speculate that the excitation of two magnetically splitted optical phonons (exchange-induced splitting of the zone-center optical phonon, see Kant *et al.*, PRL **108**, 177203 (2012)) could lead to a beating pattern at this low energy of 17 meV, which in turn modulates the 2PPE signal. However, since there are no further hints for such a (rather complicated) coupling mechanism, we did not consider this scenario. Note that the experimentally used laser fluences are rather low (as compared to laser fluences used to study phonon-driven processes; see e.g. Ref. Korff-Schmising *et al.*, Phys. Rev. B **73**, 212202 (2006)) and that the laser pump is not resonant to any phonon excitation.

3. The experimental observables occur at 3.8 eV and are interpreted in terms of the electronic structure of NiO. This is probably correct, and can be supported by the NiO film thickness dependence. The authors, however, do not emphasize how their results depend on the NiO film thickness. This is important, because 3.8 eV also corresponds

to the bulk plasmon in Ag(100) substrate,² so they could have a process, where a bulk plasmon is excited and transfers an electron to NiO.³ This certainly happens for Ag/TiO₂ interfaces. Moreover Ag has interband transitions that start at this energy.⁴ For example, Ag d-sp band transitions occur at and above this energy, and could have an influence on the NiO dynamics.

We consider the matching between the 3.8 eV onset for the excitation of the THz oscillations and the energy of the Ag substrate plasmon as accidental: (i) The Ag plasmon is a rather sharp resonance at 3.8 eV, whereas we find for the NiO dynamics an onset at 3.8 eV with an increasing excitation cross section for larger photon energies. (ii) The THz excitation appears over a wide range of NiO thicknesses with no indication for an interface-related transport mechanism. (iii) The initial ultrafast excitation (below 10 fs) that triggers the THz oscillations renders a transport mechanism through up to 20 ML NiO within this time span rather difficult. Nevertheless, we added a brief comment in the manuscript with reference to the work of Tan *et al.* [11] as indicated by the referee.

Reviewer #3 – NCOMMS-19-1124742-T

The manuscript by Gillmeister et al. describes a combined experiment and theoretical study of the photodoping and subsequent relaxation of NiO thin films. The data show long-lived excitations which live inside the gap. The description of this behavior is that of strongly correlated dynamics. The most compelling results in the data are how the decay time of the oscillations scales with the temperature.

Overall, I like the paper and lean toward recommending it for publication. However, I found much of the theory discussion and the integration of the theory with the experiment to be quite confusing. I hope this can be clarified in the next round.

We are pleased that the Referee is interested in our result. We made major revisions to address his/her suggestions. In particular, we focused on improving the connection between the theory and experiment. We agree that in the previous version there were some elements that might be misleading, like the effect of finite time resolution of the experimental probes. We hope that the Referee appreciates the revised manuscript.

1. To start, the first part of the manuscript discusses a clear charge-transfer insulator with clear upper and lower Hubbard bands and the charge-transfer band. The ensuing discussion explains how the p-holes become correlated (Zhang-Rice physics) but later we are told they are integrated out because they are noninteracting.

We thank the referee for this comment, which is indeed quite subtle. Going back to the original work of Zhang and Rice, Ref. 56, we can see that they start with strongly interacting d orbitals and noninteracting p orbitals. After a unitary transformation they arrive at a composite object involving p and d states that is effectively behaving as a hole in the spin background and which is, as stated by the referee, strongly correlated. So following the strategy of Zhang and Rice we can integrate out the p orbitals, and a proper treatment of the strong Coulomb interaction on the d orbital will lead to a correlated hole in the spin background.

However, in this work, we mainly focus on the subspace of one triplon, see the derivation in Lenarcic *et al.*, Phys.Rev. B **90**, 235136 (2014) for copper oxides, which can be easily generalized to nickel oxide. In this case the triplon is strongly interacting with the spin background, while the p orbitals are outside of the t - J description. Also in this problem, we can formally integrate out the p orbitals, resulting in a triplon in a correlated spin background.

To make these considerations more precise and avoid confusions, we reformulated the text by removing the statement on weakly interacting p orbitals and rather state “The UV pump pulse may thus be modeled within the rotating wave approximation exciting electrons from fully occupied orbitals.”

2. I found it very difficult to reconcile Figure 4 with the experimental results. How exactly am I to do that? The equilibrium spectra seemed to have far more bands and gaps than the earlier discussion described.

This is indeed an important question. Our further investigations of this point have uncovered a mistake in the calculations, which fortunately did not have a qualitative effect on the results and their physical interpretation. After fixing this issue, we have rerun all the simulations and below discuss the features of the corrected theory results.

First, let us explain the equilibrium spectra. The UHB is roughly separated from the Hund's states into three subbands (two of them visible in Fig. 4(a)) originating from the Hund's excitations, see the analysis of the equilibrium and nonequilibrium spectrum in the Supplementary Material. Due to the coupling with the AFM background, the UHB forms an additional fine structure. As explained in the revisited version, Hund's and AFM features have been observed in several other correlated materials using photoemission spectroscopy for the states below the chemical potential. For Hund's multiplets, see D. Sutter *et al.*, Nat. Comm. **8**, 15176 (2017) on Ca_2RuO_4 , and for AFM states, see G. Sangiovanni *et al.*, Phys. Rev. B **73**, 205121 (2006) for experiments on V_2O_3 . Both experiments have been compared with the DMFT description for the unoccupied states. Bała, Oleś and Zaanen computed the fine AFM structure for NiO using a parametrized approach (see figure 2 below). In our work, we have extended this analysis for excited states as now explicitly stated in the main text. The fine structure is rather weak in absolute units as we do not plot the momentum resolved spectrum. Therefore, a log-scale is employed to emphasize the presence of the photo-induced in-gap states. The current resolution of the experiment does not allow for the clear distinction of these fine features and in equilibrium we have only focused on the position of the charge-transfer gap.

In nonequilibrium situation, the experiment shows one in-gap peak at 1.2 eV. However, in theory, we see several in-gap peaks and we need to understand what is the origin of this discrepancy. Our theoretical analysis predicts a peak at $\omega \approx 0.5$, which cannot be observed in experiment due to the employed photon energy that gives the cut-off at 1 eV. In the theoretical spectra, there are three additional peaks after the photo-excitation at $\omega \approx 1.5, 2.5$, and 3.5 . In the experiment, the signal at these energies is present for very short times and then it disappears. We do not understand the origin of this phenomenon and we leave it as an open question for future studies. In practice, we have identified the experimental peak with the sideband marked by $3J_H$ in the Fig. 4(b).

We have substantially modified the text to further describe our findings. In particular, we have added the following information:

1. In the calculated equilibrium spectral function (black line in Fig. 4(a)), subbands of the UHB at energies 4.5 and 6 eV can be identified. These dominant peaks are related to Hund excitations.
2. In addition, we can identify small sidebands which originate from the coupling of charge carriers to the AFM background and the presence of string states.
3. In the Supplementary Material, we have added a new section which explains the electron insertion and removal processes associated with the features seen in the experimental spectra.
4. This analysis allows us to identify the dominant feature in the experiments, shown in Fig. 2(a), as the Hund excitation $S0^b$.

3. The nonequilibrium result had additional structure, but I was confused by this because I thought it was shown after the pump is off. By that point, the retarded Green's function should have returned to its equilibrium form because it rapidly does this (due to the fact that it represents the available quantum states and there is no dressing when the field is off). But this one has additional structure (I am guessing due to the occupancy in the UHB, but the explanation for this was not so clear).

Figure 2: Electron spectral function for the realistic parameters of NiO at the Y-point for $J_{\text{ex}} = 0.1$ (full), 0.2 (dashed) and 0.5 eV (dotted) exchange interaction, adapted from Bała, Oleś and Zaanen, Phys. Rev. B **61**, 13573 (2000).

We thank the referee for raising this question, as it is indeed important to discriminate several possibilities for the additional structure in the transient spectral function. The laser pulse is on for only a very short amount of time $t < 2$ fs, but the oscillations persist for much longer times and indicate the presence of triplons. This rules out field-induced effects. After the pulse, due to the photo-induced population of long-lived new quantum states, and the availability of new scattering channels, the spectrum exhibits long-lived nonthermal signatures. This is our main theoretical message.

As the referee has pointed out, the presence of the in-gap peak originates from the long-lived triplon states. The triplons are created by the initial pump pulse. In the photoemission process an electron is removed from the system, see Fig. 3 below. Removal of an electron from the triplon can either lead to the ground state S1 (first row) or an excited Hund's state (second row). If the final state is the Hund's state this will lead to a sideband in the PES spectrum. We have reformulated the discussion in the main text and added a section in the Supplementary Material (Section NONEQUILIBRIUM SPECTRAL FUNCTION AND RELAXATION MECHANISMS), which explains the theoretical picture in detail.

Figure 3: The photoemission process corresponds to the removal of an electron from the excited system. Due to the presence of the triplon, the system can either end up in the ground state (first row) or in an excited Hund state (second row). The latter process leads to the in-gap signal around 1 eV that exhibits long-time coherent oscillations.

4. The oscillations shown in Fig. 4 were very rapid. Indeed, they would never be seen in a real experiment because the probes have short time width envelopes, but this physics is not included in the calculations if I followed them properly.

We appreciate this important comment. It stimulated us to perform some additional data

analysis and to consider the effect of the probe envelope.

Our original idea to connect the state of the art simulations with the experimental observations was to artificially enhance the value of the superexchange and then see how the oscillations scale as we reduce the superexchange parameter J_{ex} . Indeed, after the photo-excitation we have observed slow oscillations which scale with the super-exchange parameter, similar to the experiments, see Fig. 4(d). While the realistic values of the super exchange are unreachable, we clearly show that the frequency of the oscillations scales with the reduced superexchange J_{ex} . In this sense, the theoretical and experimental results are consistent.

In the new version, we have incorporated the finite time (energy) width of the probe envelope. This is realized by the application of a low-pass filter. The threshold is determined by matching the temporal resolution of the TR-2PPE experiment (properly rescaled to take into account a much larger value of J_{ex} due to the numerical challenges of long-time propagation). We now show two different curves in Fig. 4(d) and 4(e): (i) the slow oscillations that scale with the magnitude of the superexchange and are directly relevant for the experiments, (ii) fast oscillations originating from the higher harmonics of the superexchange scale and Hund's excitations [thin lines]. The same analysis has been done for the kinetic energy.

The fast oscillations associated with the Hund's physics are also an important finding. As the time resolution of the experimental techniques is improving, they might become observable. In anticipation of future experimental studies, we hence show this theoretical prediction [thin lines] of fast oscillations originating from the Hund's excitations.

5. So, for me, there seems to be a huge disconnect between the actual experiments and the theory that is supposed to illustrate the phenomena seen in the experiment. I hope this can be properly remedied, as I did like the paper on the whole. My recommendation is to reconsider after the manuscript is revised.

While the results of the theoretical model and the experiments do not agree quantitatively, there is a qualitative agreement. In the revised version, we have also pointed out several possible technical directions that could resolve this issue, see Theoretical Methods. The proposed theoretical model can explain the important properties listed in the following table, and provide relevant physical insights:

Observations Experiment**Theory**

In-gap peak	A new in-gap peak appearing after the initial pump in the photoemission spectra.	Appearance of an in-gap peak after photo-excitation, which is identified with the photo-excited Hund state. This peak can be seen in the photoemission spectroscopy due to presence of triplons. The energy scales for the in-gap peak match in the experiment and theory, where the input comes from the ab-initio studies.
Frequency of oscillations	The intensity of the in-gap peak in the photoemission shows long-time coherence and oscillations with a frequency close to the superexchange energy.	We observe slow oscillations in the intensity of the in-gap peak that scale with the superexchange J_{ex} .
Lifetime of oscillations	The lifetime of the oscillations in the experiment is temperature dependent and is reduced to zero slightly above the Néel temperature.	We have explicitly shown that the oscillations vanish at the Néel temperature. The current theoretical description cannot capture short-range fluctuations, which is explicitly stated as a possible extension.

We thank the referee for the careful reading of our manuscript. His/her criticism has helped us to improve the discussion of the connection between the theoretical and experimental part of the paper.

List of changes

1. Abstract: their lifetime vanishes slightly above the Néel temperature.
2. Abstract: antiferromagnetic spin correlations
3. Page 2[first paragraph]: represents
4. Page 2-4, text reordered and partially modified
5. A text explaining the lattice and the spin structure of NiO is added in paragraph b. *Two-band t - J model* before Eq.(1). The passage contains a reference to the experimental paper of Fiebig *et al.* where a detailed account of these structures is given and a original paper of Anderson on the kinetic superexchange mechanism.
6. Page 4, removed text “a holon state”
7. Page 4, Discussion of assignment of photoemission features C and S with comparison to scanning tunneling spectroscopy data added.
8. Page 6-7, Discussion of Fig. 3b-e modified, which includes the separation between the thin-film Néel temperature and the critical temperature for magnetic short-range correlations.
9. Fig. 3b-e, THz oscillation lifetime show separately for two different NiO thicknesses. Data for magnetic scattering shown in addition. Fitted temperature dependencies with critical exponents from neutron scattering.
10. Page 6, Added text “We argue that the slightly larger value of T' is due to short-range spin correlations, which persist above T_N and influence the time-resolved photoemission experiments. A similar persistence of short-range magnetic order above the Curie temperature has been observed in ferromagnetic iron ²³.”
11. Page 6, A comment about plasmonic coupling across the substrate-NiO interface is added.
12. Fig. 4. Substantial modifications in the figures and captions as discussed in the reply above.
13. Fig. 5a. Corrections $J_1 \rightarrow J_2$ and $J_{ex} \rightarrow zJ_{ex}$
14. Page 8-9, Equations 1 and 2 and their discussion moved to the section ‘Theoretical Methods’.
15. Page 8, text modified/added “In addition, we can identify small sidebands which originate from the coupling of charge carriers to the AFM background and the presence of string states ³³⁻³⁷.”
16. Page 10, changed discussion “As the fast Hund oscillations cannot be resolved in the experiment, we apply a low-pass filter that extracts the slow oscillations associated with the low-energy peak (see Methods for details). The slow oscillations are shown by the black line and are found to scale with J_{ex} . We thus attribute their origin to the AFM spin correlations in NiO. To further support this interpretation, we compare the time evolution of the kinetic energy below and at T_N in Fig. 4(e).”
17. Page 10, Description and discussion of Fig. 4 modified according to reply discussed above.

18. Page 12, "The experimental observation of oscillations slightly above T_N cannot be explained by our theoretical treatment, which relies on a mean-field decoupling of the spin-spin interaction. It would be interesting to extend the formalism to fluctuating spins⁴⁸ in a future study."
19. Pages 16, added text "The UV pump pulse may thus be modeled within the rotating wave approximation exciting electrons from fully occupied orbitals."
20. Added references:
 - M. Pickel, A. Schmidt, M. Weinelt, and M. Donath, Phys. Rev. Lett. **104**, 237204 (2010).
 - S. Tan, A. Argondizzo, J. Ren, L. Liu, J. Zhao, and H. Petek, Nature Photonics **11**, 806 (2017).
 - D. Sutter et al., Nature Communications **8**, 1 (2017).
 - N. Bittner, D. Golež, M. Eckstein, and P. Werner, Phys. Rev. B **101**, 085127 (2020).
 - P. W. Anderson, Phys. Rev. **115**, 2 (1959).
 - Z. Lenarcic and P. Prelovsek, Phys. Rev. Lett. **111**, 016401 (2013)
 - Z. Lenarcic and P. Prelovsek, Phys. Rev. B **90**, 235136 (2014)
 - G. Sangiovanni, A. Toschi, E. Koch, K. Held, M. Capone, C. Castellani, O. Gunnarsson, S.-K. Mo, J. Allen, H.-D. Kim *et al.*, Phys. Rev. B **73**, 205121 (2006)

References

- [1] M. Fiebig, D. Fröhlich, T. Lottermoser, V. V. Pavlov, R. V. Pisarev, and H.-J. Weber, "Second Harmonic Generation in the Centrosymmetric Antiferromagnet NiO," Phys. Rev. Lett. **87**, 137202 (2001).
- [2] R. E. Dietz, G. I. Parisot, and A. E. Meixner, "Infrared Absorption and Raman Scattering by Two-Magnon Processes in NiO," Phys. Rev. B **4**, 2302 (1971).
- [3] M. T. Hutchings and E. J. Samuelsen, "Measurement of Spin-Wave Dispersion in NiO by Inelastic Neutron Scattering and Its Relation to Magnetic Properties," Phys. Rev. B **6**, 3447 (1972).
- [4] R. Strack and D. Vollhardt, "Dynamics of a hole in the $t - J$ model with local disorder: Exact results for high dimensions," Phys. Rev. B **46**, 13852 (1992).
- [5] R. Tuovinen, D. Golež, M. Schüler, P. Werner, M. Eckstein, and M. A. Sentef, "Adiabatic preparation of a correlated symmetry-broken initial state with the generalized kadanoff–baym ansatz," Phys. Stat. Solidi B **256**, 1800469 (2019).
- [6] P. Lipavský, V. Špička, and B. Velický, "Generalized Kadanoff-Baym ansatz for deriving quantum transport equations," Phys. Rev. B **34**, 6933 (1986).
- [7] M. Schüler, M. Eckstein, and P. Werner, "Truncating the memory time in nonequilibrium dynamical mean field theory calculations," Phys. Rev. B **97**, 245129 (2018).
- [8] T. Chatterji, G. J. McIntyre, and P.-A. Lindgard, "Antiferromagnetic phase transition and spin correlations in nio," Phys. Rev. B **79**, 172403 (2009).
- [9] M. Pickel, A. Schmidt, M. Weinelt, and M. Donath, "Magnetic exchange splitting in fe above the curie temperature," Phys. Rev. Lett. **104**, 237204 (2010).

- [10] N. Bittner, D. Golež, H. U. R. Strand, M. Eckstein, and P. Werner, “Coupled charge and spin dynamics in a photoexcited doped Mott insulator,” *Phys. Rev. B* **97**, 235125 (2018).
- [11] S. Tan, A. Argondizzo, J. Ren, L. Liu, J. Zhao, and H. Petek, “Plasmonic coupling at a metal/semiconductor interface,” *Nature Photonics* **11**, 806 (2017).

Reviewers' Comments:

Reviewer #1:

Remarks to the Author:

The authors have provided very good quality answers to all the points I raised. I consider the work to be an important contribution to ultrafast dynamics in correlated materials and support publication.

I might even go so far as to say that the manuscript would be suitable for a higher impact journal.

Reviewer #2:

Remarks to the Author:

I have read the revised manuscript by Gillmeister et al. carefully, and unfortunately I find it lacking. The authors perform a dynamical study of some presumably spin dependent phenomenon in NiO, but for the most part their experimental measurement and theoretical interpretation lacks an intelligible dynamical interpretation. Also, the author's response to my earlier comments shows a lack of understanding of relatively well understood dynamical phenomena. For this reason, I do not believe that the article can be published in the present form.

The authors excite the sample with photon energy >3.8 eV, and probe photoemission with a longer wavelength light (3.4 eV in fig. 2). They observe a 17 THz oscillation that appears to have a sine phase (Fig. 3), and persists for ~ 2 ps. The 2PPE spectroscopy of NiO has not been established, or at least the authors do not cite any relevant work. The excitation is said to occur into UHB, followed by relaxation into a photoinduced-ingap state (PIS) on 10 fs. The reader is never told whether this PIS is known from any other studies or theory. My guess is that doping the material should populate this state, and it should be known for this material. It presumably does not have to be invented to explain the results. Also, why should we believe that the relaxation from UHB to PIS occurs on 10 fs time scale? Have electron lifetimes been measured for this material? Is there related research that would support such fast relaxation?

Following formation of PIS, the authors see an oscillation of unclear origin. They present an elaborate theory claiming electron correlation effects, but as far as I can tell, this is purely a ground state model that does not explain why this process occurs. For example, why is the probe pulse sensitive to the proposed dynamics? Is photoemission from one spin-configuration much more probable than another? Why? Does it depend on the detection energy? Why does it occur with approximately the sine phase, and can one learn something from this.

The authors show a very nice photoelectron energy vs. delay time plot in Fig. 2a, and then they do not do anything with it other than to show that the high energy electron decay is fast. The phase of oscillations, actually, looks to be cos in this figure. Sin and cos matters, because it tells what kind of force induces the oscillations, but that does not seem to trouble the authors. Moreover, the oscillation appears not to be single frequency, though the data in Fig. 3, which are processed differently, do appear to follow a single damped sine behavior. I would expect that additional dynamics should appear in the two-dimensional plot in figure 2 because the carrier density and temperature are certainly varying over the 2 ps time period. But the authors do not make use of such information. To me this is a clear lack of thinking about what is responsible for the observed dynamics.

Next, I will comment on the author's response to my first comments, and some other aspects that I find troubling. My point in referring to the work of Shang¹ is that photoexcitation of NiO is likely to change the occupation of t_{2g} and e_g orbitals, and thereby exert a force on the lattice. This will happen no matter what the laser fluence may be. Furthermore, the authors claim that the oscillation that they observe is linked to the Neel temperature of the lattice. That may be true for

the initial state of the system, but once they excite the system, the electron and lattice temperature will be delay dependent. The authors do not discuss these temperatures, and their analysis is appropriate for the ground state. If the ground state Neel temperature is important for this experiment, it need not have been done in the time domain. The authors should at least present the pump power dependence of the oscillation to show that they are measuring a time-dependent phenomenon of the electronic system. Since the authors are studying a charge transfer insulator, one would expect that the charge, lattice, and spin degrees are strongly coupled.

Next, concerning the possibility of the longitudinal plasmon in Ag being involved, the authors seem to have a concept of the plasmonic response that is not in the publication that I mentioned,² and seems to be more in tune with popular literature. The authors should refer to Fig. 7 of the publication by Barman et al.³, which shows that the longitudinal plasmon excitation of Ag can inject electrons from the substrate into their 10-20 ML thick film. Assuming that these electrons have the Fermi velocity of ~ 1.5 nm/fs, it should not take very long time to be transported through their NiO film. The authors should show the pump photon energy dependence of their signal to convince this referee, that they are driving this process by exciting UHP internally rather than by excitation of the substrate.

Finally, I should mention that I find the first paragraph of the discussion on page 10 particularly disappointing. The authors take several examples of coherent dynamics that happen in vacuum, and claim that theirs is particularly novel because they observe the dynamics in a solid. It seems that they are not aware of much work on coherent phenomena in solids that occurs on comparable time scales and involves spin, charge, and lattice degrees of freedom.⁴⁻⁶ They need to compare their results to relevant phenomena, if they want to make a statement about significance of their work.

1 Shang, H. et al. Electron-phonon coupling in d-electron solids: A temperature-dependent study of rutile TiO₂ by first-principles theory and two-photon photoemission. *Physical Review Research* 1, 033153, doi:10.1103/PhysRevResearch.1.033153 (2019).

2 Reutzl, M., Li, A., Gumhalter, B. & Petek, H. Nonlinear Plasmonic Photoelectron Response of Ag(111). *Phys. Rev. Lett.* 123, 017404, doi:10.1103/PhysRevLett.123.017404 (2019).

3 Barman, S. R., Biswas, C. & Horn, K. Electronic excitations on silver surfaces. *Phys. Rev. B* 69, 045413 (2004).

4 Bovensiepen, U. Coherent and incoherent excitations of the Gd(0001) surface on ultrafast timescales. *J. Phys.: Condens. Matter* 19, 083201 (2007).

5 Bao, J., Bragas, A. V., Furdyna, J. K. & Merlin, R. Optically induced multispin entanglement in a semiconductor quantum well. *Nat Mater* 2, 175-179 (2003).

6 Winkelmann, A., Lin, W.-C., Bisio, F., Petek, H. & Kirschner, J. Interferometric Control of Spin-Polarized Electron Populations at a Metal Surface Observed by Multiphoton Photoemission. *Phys. Rev. Lett.* 100, 206601-206604 (2008).

Reviewer #3:

Remarks to the Author:

I am happy with the changes and explanations made by the authors in this revised version of the manuscript. I now recommend it be accepted for publication.

We thank all three Referees for their careful review. Whereas Reviewer #1 and #3 support strongly the publication of our revised manuscript without any further changes, Reviewer #2 raised new questions and criticism. This criticism can be refuted as outlined below.

Reviewer #1 – NCOMMS-19-1124742-T

The authors have provided very good quality answers to all the points I raised. I consider the work to be an important contribution to ultrafast dynamics in correlated materials and support publication. I might even go so far as to say that the manuscript would be suitable for a higher impact journal.

Reviewer #3 – NCOMMS-19-1124742-T

I am happy with the changes and explanations made by the authors in this revised version of the manuscript. I now recommend it be accepted for publication.

Reviewer #2 – NCOMMS-19-1124742-T

I have read the revised manuscript by Gillmeister et al. carefully, and unfortunately I find it lacking. The authors perform a dynamical study of some presumably spin dependent phenomenon in NiO, but for the most part their experimental measurement and theoretical interpretation lacks an intelligible dynamical interpretation. Also, the author's response to my earlier comments shows a lack of understanding of relatively well understood dynamical phenomena. For this reason, I do not believe that the article can be published in the present form.

We respectfully disagree with this overall judgement, which is inconsistent with the previous report of the same Referee:

Referee #2 on initial submission: Gillmeister et al perform ultrafast 2PPE experiment on NiO. They observe coherent THz oscillations, which they interpret in terms of Hund's physics of spin interactions associated with electron injection into the eg symmetry orbitals. The experiment and theoretical analysis are well done, so I recommend publication. Such physics have not been observed in the time domain, as far as I am aware, and I believe it will be of broad interest to those investigating correlated materials in the time domain. It is quite sophisticated that the experiments are accompanied by high level theory. The authors may consider a few comments.

In particular, we are surprised that the new questions mainly refer to topics which were not raised in the first round of reviewing. However, we welcome the opportunity to clarify these points. Please find our detailed point-by-point responses following the copy of the Referee's comments below.

1. The authors excite the sample with photon energy >3.8 eV, and probe photoemission with a longer wavelength light (3.4 eV in fig. 2). They observe a 17 THz oscillation that appears to have a sine phase (Fig. 3), and persists for 2 ps. The 2PPE spectroscopy of NiO has not been established, or at least the authors do not cite any relevant work. The excitation is said to occur into UHB, followed by relaxation into a photoinduced-ingap state (PIS) on 10 fs. The reader is never told whether this PIS is known from any other studies or theory. My guess is that doping the material should populate this state, and it should be known for this material. It presumably does not have to be invented to explain the results. Also, why should we believe that the relaxation from UHB to PIS occurs on 10 fs time scale? Have electron lifetimes been measured for this material? Is there related research that would support such fast relaxation?

Small clarification: The THz oscillations are observed at a frequency of 4.2 and not at 17 THz as stated by the Referee. This is important because the frequency of 17 THz would correspond to the LO phonon mode in NiO [1].

The underlying stationary electronic states are well-known since the pioneering works of Zhang and Rice [2] and Bała *et al.* [3]. The doped system has just recently attracted theoretical attention, both for hole [4] and electron doping [5]. These equilibrium DMFT plus *ab initio* DFT band structure calculations report that “for both, electron and hole doping, there appears a narrow band of occupied states at the top of the valence band, which is more pronounced for hole than for electron doping” (as can be seen in a very illustrative Fig. 12 of Ref. [5]). The novelty of our work is the analysis of the photo-doped state and the observation and explanation of coherent photo-induced effects. These effects are generically not present in bulk systems (fast decoherence), but we developed a many-body theory that elucidates the microscopic mechanism.

To the best of our knowledge, this is the first observation of many-body coherent oscillations in transition metal oxides. We agree that there are several other setups that show similar effects: a) the setups mentioned in point 6 of the referee, b) recent experiments on transition-metal dichalcogenides (strong candidates for excitonic insulators, see Ref. [6, 7]), where the coupling between phonons and collective modes has been proposed to explain long-time coherence, c) polaronic effects in materials like 1T-TaS₂ [8]. These examples suggest that coherent dynamics can originate from several different microscopic mechanisms. Our combined theoretical and experimental analysis has revealed an unanticipated microscopic origin and proposed a class of materials, namely multi-band insulators with Hund coupling (nickelates, ruthenates), where long-lived coherent phenomena can be observed. Moreover, the theoretical analysis yielded a clear suggestion for analogous cold-atoms experiments. While in the previous version we have focused on cold-atom systems, we have modified this sentence in the resubmitted version: “Various solid state systems exhibit coherent oscillations, which can emerge for impurity [9] and surface states [10, 11] or due to the strong coupling to collective [6, 7] and phononic modes [8].”

The ultrafast relaxation (≤ 10 fs) of the initial electronic excitation in the UHB is shown in the time-resolved data of Fig. 2a and b, as well as Fig. 4 in the Supplemental Information for NiO film thicknesses between 4 and 20 ML. From the theory point of view, this is expected in strongly correlated multi-orbital systems due to the large Hund coupling (≈ 1 eV), as stated in the main text. This effect has been explicitly evaluated in the many-body simulations and it agrees with the estimation of 10 fs from the experiment. Moreover, the fast relaxation has been previously reported in Ref. [12], which is now added in the main text. For experiments, the current time resolution is not short enough to monitor this dynamics, resolving it represents an important goal for future studies.

2. Following formation of PIS, the authors see an oscillation of unclear origin. They present an elaborate theory claiming electron correlation effects, but as far as I can tell, this is purely a ground state model that does not explain why this process occurs. For example, why is the probe pulse sensitive to the proposed dynamics? Is photoemission from one spin-configuration much more probable than another? Why? Does it depend on the detection energy? Why does it occur with approximately the sine phase, and can one learn something from this.

We respectfully disagree with the referee’s comment. We have revealed a detailed and intuitive microscopic picture for the oscillations and supported the minimal modeling with ex-

PLICIT simulations. As stated in the method part, we perform our theoretical analysis by using the *full* Keldysh description without any assumptions of the ground state.

We agree that before this work it was not clear why the probe pulse would be sensitive to the proposed dynamics. One of the main insights from this study is the sensitivity of the photo-excited in-gap state on the probe-pulse photoemission. The relevant mechanism is described in half a paragraph of the main text [page 8, starting with: “In the experiments, the photoexcited system ...”]. Furthermore, as the referee correctly points out, the matrix elements describing the coupling to the probe field are not included in the calculations of the photoemission spectrum in Eq. (7). This would require a fully *ab initio* approach that is currently impossible. However, we expect that oscillations would be even more pronounced if the procedure is fully implemented. We do mention here the two latest theoretical works on this system [4, 5]. They illustrate the difficulties one faces already in the equilibrium state. The nonequilibrium *ab initio* treatment goes far beyond the scope of the current paper and the state-of-the-art in the field.

A strongly time-dependent electron temperature, phonon temperature, or high carrier density, as present in many time-resolved ARPES experiments with higher laser pump fluences, is not essential in the 2PPE experiments reported here, where we are dealing with a low density of excitations (not the ground state regime). The experimentally observed oscillations are present over the technically accessible range of probe photon energies as demonstrated in Fig. 5 of the Supplemental Information and we do not find variations in the oscillation frequency, the decay rate, or the phase. Similarly, our theoretical treatment is appropriate for exactly this low excitation density regime (which is different from the ground state). In this regime, AFM correlations are not destroyed.

3. The authors show a very nice photoelectron energy vs. delay time plot in Fig. 2a, and then they do not do anything with it other than to show that the high energy electron decay is fast. The phase of oscillations, actually, looks to be cos in this figure. Sin and cos matters, because it tells what kind of force induces the oscillations, but that does not seem to trouble the authors. Moreover, the oscillation appears not to be single frequency, though the data in Fig. 3, which are processed differently, do appear to follow a single damped sine behavior. I would expect that additional dynamics should appear in the two-dimensional plot in figure 2 because the carrier density and temperature are certainly varying over the 2 ps time period. But the authors do not make use of such information. To me this is a clear lack of thinking about what is responsible for the observed dynamics.

We thank the Referee for his/her useful insights.

The data exemplified in Fig. 2a on a time scale of 3 ps are analyzed and discussed in more details in the context of Fig. 3. The data of Fig. 2a are included in Fig. 3a as the trace for 150 K (blue dots), where in Fig. 3a a slowly decaying background is subtracted for all spectra (as mentioned in the figure caption). A close inspection of Fig. 3a indicates that the first maximum is at slightly positive delay times, independent of sample temperature. In fact, as the fits of the 2PPE data in Fig. 3a clearly demonstrate, the oscillations follow a (single-frequency) damped cosine-like form as stated in the manuscript. To make it unambiguous, we now added the value of the fitting parameter $\phi \approx 0$ on page 5 of the revised manuscript.

While we have not elaborated on the cos/sin question in the theory discussion, the dispersive mechanism is well compatible with our theoretical modeling. In the main text, we explicitly stated that the dynamics of the photo-doped charges in an AFM lattice can be understood as that of a particle in an effective linear potential. The pump pulse creates long FM defects (string states) in the AFM background, which are equivalent to a fast displacement in the

effective potential. An approximate cosine response is a natural consequence of this. We have added the following explanation in the main text: “The string of FM distortions is rapidly created by the pump pulse, which acts as a displacive excitation within the string potential. A natural consequence is a cosine-like response as evident in the experimental data, see Fig. 2, and the theoretical response, see Fig. 4.”

We have already addressed the question of varying carrier density and temperature in point 2 by emphasizing the low photo-doping regime.

4. Next, I will comment on the author’s response to my first comments, and some other aspects that I find troubling. My point in referring to the work of Shang [13] is that photoexcitation of NiO is likely to change the occupation of t2g and eg orbitals, and thereby exert a force on the lattice. This will happen no matter what the laser fluence may be. Furthermore, the authors claim that the oscillation that they observe is linked to the Neel temperature of the lattice. That may be true for the initial state of the system, but once they excite the system, the electron and lattice temperature will be delay dependent. The authors do not discuss these temperatures, and their analysis is appropriate for the ground state. If the ground state Neel temperature is important for this experiment, it need not have been done in the time domain. The authors should at least present the pump power dependence of the oscillation to show that they are measuring a time-dependent phenomenon of the electronic system. Since the authors are studying a charge transfer insulator, one would expect that the charge, lattice, and spin degrees are strongly coupled.

We did take the comments from the first round seriously, and in order to eliminate other possibilities such as plasmons and phonons, new temperature-dependent data showing a threshold behavior at T_N have been presented and discussed (Fig. 3). The dependence on the pump fluence (Fig. 6 of Supplemental Information) and the photon energy (Fig. 3e) are likewise presented and discussed.

As we have already pointed out in the reply to question 2, our experiments are performed in the low photo-doping regime (not the ground state). On the experimental side, this means that the photo-induced signal is increasing linearly with the pump fluence, the photo-induced excitations are well separated, and we are far from destroying AFM correlations in the system. The theoretical modeling does not rely on a ground state assumption, but correlations present in the initial state have a significant effect on the dynamics.

5. Next, concerning the possibility of the longitudinal plasmon in Ag being involved, the authors seem to have a concept of the plasmonic response that is not in the publication that I mentioned [14], and seems to be more in tune with popular literature. The authors should refer to Fig. 7 of the publication by Barman et al. [15], which shows that the longitudinal plasmon excitation of Ag can inject electrons from the substrate into their 10-20 ML thick film. Assuming that these electrons have the Fermi velocity of 1.5 nm/fs, it should not take very long time to be transported through their NiO film. The authors should show the pump photon energy dependence of their signal to convince this referee, that they are driving this process by exciting UHP internally rather than by excitation of the substrate.

We thank the referee for these remarks on an alternative excitation of the THz oscillations within the NiO spin system. Indeed, we considered but disregarded this pathway based on the following arguments. (i) It is true that silver has a strong bulk plasmon and related surface monopole and weaker multipole surface plasmons that might interact with time-varying electric fields. At a planar interface (as in our case), however, the coupling between incident

light and the strong surface monopole plasmon is forbidden by symmetry. The experimental work by Barman'04 mentioned by the referee studied the photoyield for silver surfaces and photon energies across the surface plasmon resonance. They find a moderate photoyield enhancement for the single crystal surface of about a factor of two, which originates from the surface plasmon mode (most likely the multipole mode). Therefore, one would expect a similar resonant enhancement for photo-induced processes that originate at the silver surface. In contrast, we find in our study a strong threshold behavior (Fig. 3(e)) where the photoemission signal changes by more than one order of magnitude. (ii) For electrons that are excited in the silver substrate, the transport to the NiO surface is hampered by the impedance mismatch due to the different band structures of Ag and NiO: The different energy-dependent group velocities in both materials lead to strong backscattering at the interface. (iii) The electrons, which are not back-scattered at the interface, might travel within the NiO conduction band to the surface. The Referee assumed for this process a group velocity of 1.5 nm/fs, which corresponds to the Fermi velocity in the free-electron-like sp bands of Ag or Au. However, at the bottom of the NiO conduction band (bottom of the upper Hubbard band) transport cannot be described based on a free-electron-like band dispersion due to the more localized d-band electron character and, of course, due to strong localization based on electron-electron correlation. An estimation of the group velocity at the bottom of the conduction band is rather difficult, but leads to an at least one-order-of-magnitude smaller group velocity and therefore to much longer transport times, which are not compatible with the experimentally determined ultrafast (<10 fs) lifetime of the initial excitation (see Fig. 2 and Supplemental Information).

Based on these three arguments (only small plasmonic photoyield enhancement, impedance mismatch, slow transport) we excluded such substrate-induced scenarios for our main observations.

6. Finally, I should mention that I find the first paragraph of the discussion on page 10 particularly disappointing. The authors take several examples of coherent dynamics that happen in vacuum, and claim that theirs is particularly novel because they observe the dynamics in a solid. It seems that they are not aware of much work on coherent phenomena in solids that occurs on comparable time scales and involves spin, charge, and lattice degrees of freedom [16–18]. They need to compare their results to relevant phenomena, if they want to make a statement about significance of their work.

We would like to emphasize that the first paragraph of the discussion on page 10 was not changed in the resubmission and not criticized by any of the referees before.

We are aware of the related work on coherent phenomena in solids. These experiments differ in important details and we have already mentioned some additional studies of the coupled dynamics in point 1. We have added additional explanations in the resubmitted manuscript.

We would like to thank Referee #2 for his/her careful reading of our manuscript, and the valuable suggestions, which we hope have been adequately addressed.

List of changes

1. Page 6, first line: The phase shift is explicitly specified.
2. Page 10, line 18: We added reference [39] as Strand *et al.*, Phys. Rev. B **96**, 165104 (2017).
3. Page 10, 4th last line: We added references to related work in the field of solid state dynamics: "Various solid state systems exhibit coherent oscillations ... phononic modes." with references to Bao *et al.*, Nat. Materials **2**, 175 (2003); Bovensiepen, J. Physics: Cond. Matt. **19**, 083201 (2007); Winkelmann *et al.*, Phys. Rev. Lett. **100**, 206601 (2008); Hellmann *et al.*, Nat. Commun. **3**, 1069 (2012); Werdehausen *et al.*, Science Adv. **4**, eaap8652 (2018); Perfetti *et al.*, New J. Phys. **10**, 053019 (2008).
4. Page 12, line 11: We added two sentences regarding the cosine-like temporal response: "A string of FM distortions is rapidly created by the pump pulse, which acts as a dispersive excitation within the string potential. A natural consequence is a cosine-like response, as evident in the experimental data (Fig. 3), and the simulation results (Fig. 4)."

References

- [1] Y. Wang, J. E. Saal, J.-J. Wang, A. Saengdeejing, S.-L. Shang, L.-Q. Chen, and Z.-K. Liu, "Broken symmetry, strong correlation, and splitting between longitudinal and transverse optical phonons of mno and nio from first principles," Phys. Rev. B **82**, 081104 (2010).
- [2] F. C. Zhang and T. M. Rice, "Effective Hamiltonian for the superconducting Cu oxides," Phys. Rev. B **37**, 3759 (1988).
- [3] J. Bała, A. M. Oleś, and J. Zaanen, "Origin of band and localized electron states in photoemission of NiO," Phys. Rev. B **61**, 13573 (2000).
- [4] F. Lechermann, W. Körner, D. F. Urban, and C. Elsässer, "Interplay of charge-transfer and Mott-Hubbard physics approached by an efficient combination of self-interaction correction and dynamical mean-field theory," Phys. Rev. B **100**, 115125 (2019).
- [5] F. Wrobel, H. Park, C. Sohn, H.-W. Hsia, J.-M. Zuo, H. Shin, H. N. Lee, P. Ganesh, A. Benali, P. R. C. Kent, O. Heinonen, and A. Bhattacharya, "Doped NiO: the Mottness of a charge transfer insulator," arXiv:2001.11436 [cond-mat] (2020).
- [6] S. Hellmann, T. Rohwer, M. Kalläne, K. Hanff, C. Sohrt, A. Stange, A. Carr, M. Murnane, H. Kapteyn, L. Kipp, M. Bauer, and K. Rossnagel, "Time-domain classification of charge-density-wave insulators," Nat. Commun. **3**, 1069 (2012).
- [7] D. Werdehausen, T. Takayama, M. Höppner, G. Albrecht, A. W. Rost, Y. Lu, D. Manske, H. Takagi, and S. Kaiser, "Coherent order parameter oscillations in the ground state of the excitonic aaaa insulator Ta₂NiSe₅," Science Advances **4**, eaap8652 (2018).
- [8] L. Perfetti, P. A. Loukakos, M. Lisowski, U. Bovensiepen, M. Wolf, H. Berger, S. Biermann, and A. Georges, "Femtosecond dynamics of electronic states in the mott insulator 1t-tas₂ by time resolved photoelectron spectroscopy," New J. Phys. **10**, 053019 (2008).
- [9] J. Bao, A. V. Bragas, J. K. Furdyna, and R. Merlin, "Optically induced multispin entanglement in a semiconductor quantum well," Nature Materials **2**, 175 (2003).
- [10] U. Bovensiepen, "Coherent and incoherent excitations of the Gd(0001) surface on ultrafast timescales," J. Phys. Condens. Matter **19**, 083201 (2007).

- [11] A. Winkelmann, W.-C. Lin, F. Bisio, H. Petek, and J. Kirschner, "Interferometric control of spin-polarized electron populations at a metal surface observed by multiphoton photoemission," *Phys. Rev. Lett.* **100**, 206601 (2008).
- [12] H. U. R. Strand, D. Golež, M. Eckstein, and P. Werner, "Hund's coupling driven photocarrier relaxation in the two-band Mott insulator," *Phys. Rev. B* **96**, 165104 (2017).
- [13] H. Shang, A. Argondizzo, S. Tan, J. Zhao, P. Rinke, C. Carbogno, M. Scheffler, and H. Petek, "Electron-phonon coupling in *d*-electron solids: A temperature-dependent study of rutile TiO₂ by first-principles theory and two-photon photoemission," *Phys. Rev. Res.* **1**, 033153 (2019).
- [14] M. Reutzler, A. Li, B. Gumhalter, and H. Petek, "Nonlinear Plasmonic Photoelectron Response of Ag(111)," *Phys. Rev. Lett.* **123**, 017404 (2019).
- [15] S. Barman, C. Biswas, and K. Horn, "Electronic excitations on silver surfaces," *Phys. Rev. B* **69**, 045413 (2004).
- [16] U. Bovensiepen, "Coherent and incoherent excitations of the Gd(0001) surface on ultrafast timescales," *J. Phys. Condens. Matter* **19**, 083201 (2007).
- [17] J. Bao, A. V. Bragas, J. K. Furdyna, and R. Merlin, "Optically induced multispin entanglement in a semiconductor quantum well," *Nat. Mater.* **2**, 175 (2003).
- [18] A. Winkelmann, W.-C. Lin, F. Bisio, H. Petek, and J. Kirschner, "Interferometric Control of Spin-Polarized Electron Populations at a Metal Surface Observed by Multiphoton Photoemission," *Phys. Rev. Lett.* **100**, 206601 (2008).

Reviewers' Comments:

Reviewer #2:

Remarks to the Author:

1589478662

The authors mostly dismiss my criticisms in the second review. I find this work to be cloaked in many body physics, which are not adequately explained, and as far as I can understand, are interpretations of the authors, without concrete support. The coherent oscillations are nicely measured, but there is no firm evidence that this is electron spin related. The oscillations disappear above the Neel temperature, which is consistent with spin, but experiments such as light polarization dependence are lacking. The origin of the coherence is opaque, The coherent oscillation do not occur in the photoexcites state but in a state that is only explained as a photoinduced gap state, which is populated by relaxation. It takes strong optimism to believe that relaxation occurs from the photoexcited state to generate an energetically relaxed coherent state. Furthermore, the authors should indicate the strength of the observed oscillation. Is it 1 or 100% effect? It should be evident in the reported figures. One should not have to ask such details. The authors claim that they detect coherent oscillations of a many body system for the first time, and then in the next sentence they say that similar phenomena have been detected in cuprates. I think only one of these statements can be correct. I do not want to belabor this. If the authors wish to respectfully disagree, I encourage the editor to seek further opinion.

Reviewer #4:

Remarks to the Author:

The manuscript from Gillmeister et al. reports on the ultrafast photoexcited dynamics of a charge-transfer insulator NiO studied by time-resolved two-photon photoemission experiments. The authors observed the photo-induced in gap states, of which spectral weight shows long-lived coherent THz oscillations. Combined with LDA+DMFT calculations on the basis of two-band t-J model, the oscillation is explained as that between two local many-body states consisted of S=0 Hund excitation and triplon states photo-created in an antiferromagnetic background.

The work is highly original and presents a convincing interpretation of the experimental results. It also illuminates the fundamental character on the interplay between charge and spin excitations in this class of charge transfer insulator. I think that the issues raised by reviewer 2 were also properly addressed in the rebuttal and the revised manuscript. Because of its novelty and general interest I strongly recommend publication of the manuscript in Nature Communications.

**Reviewer #4 (Remarks to the Author) –
NCOMMS-19-1124742B:**

The manuscript from Gillmeister et al. reports on the ultrafast photoexcited dynamics of a charge-transfer insulator NiO studied by time-resolved two-photon photoemission experiments. The authors observed the photo-induced in gap states, of which spectral weight shows long-lived coherent THz oscillations. Combined with LDA+DMFT calculations on the basis of two-band t - J model, the oscillation is explained as that between two local many-body states consisted of $S=0$ Hund excitation and triplon states photo-created in an antiferromagnetic background.

The work is highly original and presents a convincing interpretation of the experimental results. It also illuminates the fundamental character on the interplay between charge and spin excitations in this class of charge transfer insulator. I think that the issues raised by reviewer 2 were also properly addressed in the rebuttal and the revised manuscript. Because of its novelty and general interest I strongly recommend publication of the manuscript in Nature Communications.

We thank the Referee for the positive assessment of our work and for recommending a publication.

**Reviewer #2 (Remarks to the Author) –
NCOMMS-19-1124742B:**

The authors mostly dismiss my criticisms in the second review. I find this work to be cloaked in many body physics, which are not adequately explained, and as far as I can understand, are interpretations of the authors, without concrete support. The coherent oscillations are nicely measured, but there is no firm evidence that this is electron spin related. The oscillations disappear above the Neel temperature, which is consistent with spin, but experiments such as light polarization dependence are lacking. The origin of the coherence is opaque, The coherent oscillation do not occur in the photoexcites state but in a state that is only exppopulated by relaxation. It takes strong optimism to believe that relaxation occurs from the photoexcited state to generate an energetically relaxed coherent state. Furthermore, the authors should indicate the strength of the observed oscillation. Is it 1 or 100effect? It should be evident in the reported figures. One should not have to ask such details. The authors claim that they detect coherent oscillations of a many body system for the first time, and then in the next sentence they say that similar phenomena have been detected in cuprates. I think only one of these statements can be correct. I do not want to belabor this. If the authors wish to respectfully disagree, I encourage the editor to seek further opinion.

We respectfully disagree with this overall judgement. Firstly, following the Reviewer #4 report, we have addressed all the issues raised by Reviewer #2 in the rebuttal and the revised manuscript. Further, as mentioned in our previous reply, we have performed experimental analysis of the photo-doped state in NiO. This is supported by the minimal theoretical modeling with the explicit simulations. The experiments with the light polarization dependence go beyond the scope of our studies. Finally, there is no contradiction between our statements about the novelty of our results and similar phenomena detected in cuprates. As mentioned in our previous reply and in the main text of our manuscript, these experiments differ in the important details and show rather a short-range coherence dynamics. Moreover, their origin remains still controversial.

The strength of the THz oscillations relative to the secondary photoemission background is directly visible from the raw data in Fig. 2a.